# Genetic basis and dual adaptive role of floral pigmentation in sunflowers

Marco Todesco[1]*, Natalia Bercovich[1], Amy Kim[1], Ivana Imerovski[1], Gregory L Owens[1,2], Óscar Dorado Ruiz[1], Srinidhi V Holalu[3], Lufiani L Madilao[4], Mojtaba Jahani[1], Jean-Sébastien Légaré[1], Benjamin K Blackman[3], Loren H Rieseberg[1]*

[1]Department of Botany and Biodiversity Research Centre, University of British Columbia, Vancouver, Canada; [2]Department of Biology, University of Victoria, Victoria, Canada; [3]Department of Plant and Microbial Biology, University of California, Berkeley, United States; [4]Michael Smith Laboratory and Wine Research Centre, University of British Columbia, Vancouver, United States

*For correspondence:
mtodesco@biodiversity.ubc.ca (MT);
lriesebe@mail.ubc.ca (LHR)

**Competing interest:** The authors declare that no competing interests exist.

**Abstract** Variation in floral displays, both between and within species, has been long known to be shaped by the mutualistic interactions that plants establish with their pollinators. However, increasing evidence suggests that abiotic selection pressures influence floral diversity as well. Here, we analyse the genetic and environmental factors that underlie patterns of floral pigmentation in wild sunflowers. While sunflower inflorescences appear invariably yellow to the human eye, they display extreme diversity for patterns of ultraviolet pigmentation, which are visible to most pollinators. We show that this diversity is largely controlled by *cis*-regulatory variation affecting a single MYB transcription factor, HaMYB111, through accumulation of ultraviolet (UV)-absorbing flavonol glycosides in ligules (the 'petals' of sunflower inflorescences). Different patterns of ultraviolet pigments in flowers are strongly correlated with pollinator preferences. Furthermore, variation for floral ultraviolet patterns is associated with environmental variables, especially relative humidity, across populations of wild sunflowers. Ligules with larger ultraviolet patterns, which are found in drier environments, show increased resistance to desiccation, suggesting a role in reducing water loss. The dual role of floral UV patterns in pollinator attraction and abiotic response reveals the complex adaptive balance underlying the evolution of floral traits.

## Editor's evaluation

The enlarged petals of sunflowers contain pigments that absorb ultraviolet light and are perceived by pollinators as dark 'bullseyes' that function as nectar guides. Todesco et al. identify the primary genetic mechanism underlying variation in the size of this bullseye pattern and provide evidence suggesting that abiotic variables, rather than pollinators, may maintain this phenotypic and genotypic variation.

## Introduction

The diversity in colour and colour patterns found in flowers is one of the most extraordinary examples of adaptive variation in the plant world. As remarkable as the variation that we can observe is, even more of it lays just outside our perception. Many species accumulate pigments that absorb ultraviolet (UV) radiation in their flowers; while these patterns are invisible to the human eye, they can be perceived by pollinators, most of which can be seen in the near UV (*Chittka et al., 1994*; *Tovée, 1995*). UV patterns have been shown to increase floral visibility and to have a major influence on

**eLife digest** Flowers are an important part of how many plants reproduce. Their distinctive colours, shapes and patterns attract specific pollinators, but they can also help to protect the plant from predators and environmental stresses.

Many flowers contain pigments that absorb ultraviolet (UV) light to display distinct UV patterns – although invisible to the human eye, most pollinators are able to see them. For example, when seen in UV, sunflowers feature a 'bullseye' with a dark centre surrounded by a reflective outer ring. The sizes and thicknesses of these rings vary a lot within and between flower species, and so far, it has been unclear what causes this variation and how it affects the plants.

To find out more, Todesco et al. studied the UV patterns in various wild sunflowers across North America by considering the ecology and molecular biology of different plants. This revealed great variation between the UV patterns of the different sunflower populations. Moreover, Todesco et al. found that a gene called *HaMYB111* is responsible for the diverse UV patterns in the sunflowers. This gene controls how plants make chemicals called flavonols that absorb UV light.

Flavonols also help to protect plants from damage caused by droughts and extreme temperatures. Todesco et al. showed that plants with larger bullseyes had more flavonols, attracted more pollinators, and were better at conserving water. Accordingly, these plants were found in drier locations.

This study suggests that, at least in sunflowers, UV patterns help both to attract pollinators and to control water loss. These insights could help to improve pollination – and consequently yield – in cultivated plants, and to develop plants with better resistance to extreme weather. This work also highlights the importance of combining biology on small and large scales to understand complex processes, such as adaptation and evolution.

pollinator visitation and preference (*Brock et al., 2016*; *Horth et al., 2014*; *Rae and Vamosi, 2013*; *Sheehan et al., 2016*). Besides their importance for pollinator attraction, patterns of UV-absorbing pigments in flowers have increasingly been recognized to have a role in responses to other biotic and abiotic factors, including defence against insect herbivory (*Gronquist et al., 2001*), protection against UV radiation (*Koski and Ashman, 2015*; *Koski et al., 2020*), and adaptation to different temperatures (*Koski and Ashman, 2016*; *Koski et al., 2020*).

Sunflowers are one of the most recognizable members of the Asteraceae family, which comprises circa 10% of all flowering plants (*Mandel et al., 2019*). Besides cultivated sunflower, about 50 species of wild sunflowers are found across North America. Wild sunflowers are adapted to a variety of different habitats and display a remarkable amount of phenotypic and genetic diversity, which makes them a model system for studies of adaptation, speciation, and domestication (*Bock et al., 2020*; *Heiser et al., 1969*; *Todesco et al., 2020*). In addition to being a major crop, sunflowers are also ubiquitous in popular culture, largely due to their iconic yellow inflorescences. Indeed, like many Asteraceae species, wild sunflowers have ligules (the enlarged modified petals of the outermost whorl of florets in the sunflower inflorescence) that appear of the same bright yellow colour to the human eye. However, ligules also accumulate UV-absorbing pigments at their base, while their tip reflects UV radiation (*Harborne and Smith, 1978*; *Wojtaszek and Maier, 2014*). Across the whole inflorescence, this results in a bullseye pattern, with an external UV-reflecting ring and an internal UV-absorbing ring. Considerable variation in the size of UV bullseye patterns has been observed between and within plant species (*Koski and Ashman, 2013*; *Koski and Ashman, 2016*); however, few studies have investigated the ecological factors that drive this variation or the genetic determinants that control it (*Brock et al., 2016*; *Koski and Ashman, 2015*; *Moyers et al., 2017*; *Sheehan et al., 2016*). Here, we explore the diversity of floral UV pigmentation in wild sunflowers and the genetic mechanisms and environmental factors that shape this variation.

## Results and discussion
### Floral UV patterns in wild sunflowers
A preliminary screening of 19 species of wild sunflowers, as well as cultivated sunflower, suggested that UV bullseye patterns are common across sunflower species (*Figure 1—figure supplement 1*). In

several cases, we also observed substantial within-species variation for the size of UV floral patterns. Patterns of floral UV pigmentation have been previously investigated in the silverleaf sunflower *Helianthus argophyllus*, which is endemic to Southern Texas (*Figure 1—figure supplement 1*). Limited diversity was found between individuals, but transgressive segregation was observed in mapping populations; while several QTL affecting this trait were detected, genetic mapping resolution was insufficient to identify individual causal genes (*Moyers et al., 2017*).

To better understand the function and genetic regulation of variation for floral UV pigmentation, we focused on two widespread species of annual sunflowers, *Helianthus annuus* and *Helianthus petiolaris*. *H. annuus*, the common sunflower, grows across most of North America; it is probably the most diverse of the sunflower species and is the progenitor of domesticated sunflower (*H. annuus* var. *macrocarpus*). *H. petiolaris* also has a broad distribution across North America, but prefers sandier soils. It includes two subspecies: subsp. *petiolaris*, which is common in the central plains of the United States, and subsp. *fallax*, which is found in the Southwestern USA and has repeatedly adapted to growing on sand dunes (*Heiser et al., 1969*; *Todesco et al., 2020*). Over two growing seasons, we measured floral UV patterns (as the proportion of the ligule that absorbs UV radiation, henceforth 'ligule ultraviolet proportion' [LUVp]) in 1589 *H. annuus* individuals derived from 110 distinct natural populations and 351 *H. petiolaris* individuals from 40 populations, grown in common garden

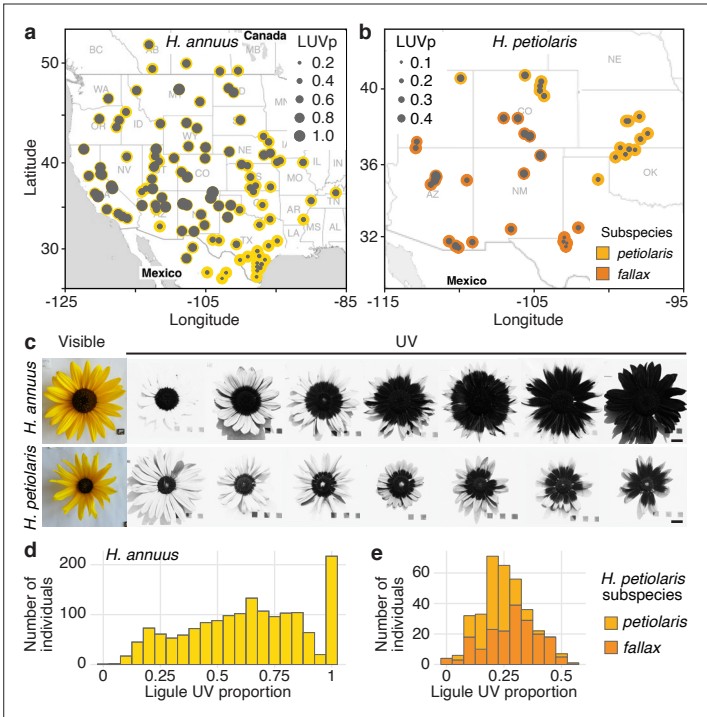

**Figure 1.** Diversity for floral ultraviolet (UV) pigmentation patterns in wild sunflowers. (**a**) Geographical distribution of sampled populations for *H. annuus* and (**b**) *H. petiolaris*. Yellow/orange dots represent different populations, overlaid grey dot size is proportional to the population mean ligule ultraviolet proportion (LUVp). (**c**) Range of variation for floral UV pigmentation patterns in the two species. Scale bar = 2 cm. (**d**) LUVp values distribution for *H. annuus* and (**e**) *H. petiolaris* subspecies.

The online version of this article includes the following source data and figure supplement(s) for figure 1:

**Source data 1.** Populations used in this study, average ligule ultraviolet proportion (LUVp) values, environmental variables, and inflorescence traits.

**Source data 2.** Individuals used in this study, ligule ultraviolet proportion (LUVp) values, Chr15_LUVp SNP genotypes, and inflorescence traits.

**Figure supplement 1.** Floral ultraviolet (UV) patterns in wild sunflower species and cultivated sunflower.

**Figure supplement 2.** Partial ultraviolet (UV) absorbance in the distal part of ligules in *H. annuus*.

**Figure supplement 3.** Ligule ultraviolet proportion (LUVp) variation in wild sunflower species and cultivated sunflower.

experiments in Vancouver, Canada (*Todesco et al., 2020*). The populations of origin of these plants were selected to represent the whole range of *H. annuus*, and most of the range of *H. petiolaris* (*Figure 1a and b*, *Figure 1—source data 1*). While extensive variation was observed within both species, it was particularly striking for *H. annuus*, which displayed a phenotypic continuum from ligules with almost no UV pigmentation to ligules that were entirely UV-absorbing (*Figure 1c–e*, *Figure 1— figure supplement 2*, *Figure 1—source data 2*). Floral UV patterns have been proposed to act as nectar guides, helping pollinators orient towards nectar rewards once they land on the petal (*Daumer, 1958*), although recent experiments have challenged this hypothesis (*Koski et al., 2014*). A relatively high proportion of *H. annuus* individuals in our survey (~13%) had completely UV-absorbing ligules and therefore lacked UV nectar guides, suggesting that pollinator orientation is not a necessary function of floral UV pigmentation in sunflower.

## Genetic control of floral UV patterning

To identify the loci controlling variation for floral UV patterning, we performed a genome-wide association study (GWAS). We used a subset of the phenotyped plants (563 of the *H. annuus* and all 351 *H. petiolaris* individuals) for which we previously generated genotypic data at >4.6M high-quality single-nucleotide polymorphisms (SNPs) (*Todesco et al., 2020*). Given their relatively high level of genetic differentiation, analyses were performed separately for the *petiolaris* and *fallax* subspecies of *H. petiolaris* (*Todesco et al., 2020*). While no significant association was identified for *H. petiolaris fallax* (*Figure 2—figure supplement 1*), we detected several genomic regions significantly associated with floral UV patterning in *H. petiolaris petiolaris*, and a particularly strong association (p=5.81e$^{-25}$) on chromosome 15 in *H. annuus* (*Figure 2a and b*). The chromosome 15 SNP with the strongest association with ligule UV pigmentation patterns in *H. annuus* (henceforth 'Chr15_LUVp SNP') explained 62% of the observed phenotypic and additive variation (narrow-sense heritability for LUVp in the *H. annuus* dataset is ~1). Additionally, allelic distributions at this SNP closely matched that of floral UV patterns (*Figure 2c*, compare to *Figure 1a*; *Figure 1—source data 2*).

Genotype at the Chr15_LUVp SNP had a remarkably strong effect on the size of UV bullseyes in inflorescences. Individuals homozygous for the 'large' (L) allele had a mean LUVp of 0.78 (SD ±0.16), meaning that ~3/4 of the ligule was UV-absorbing, while individuals homozygous for the 'small' (S) allele had a mean LUVp of 0.33 (SD ±0.15), meaning that only the basal ~1/3 of the ligule absorbed UV radiation. Consistent with the trimodal LUVp distribution observed for *H. annuus* (*Figure 1d*), alleles at this locus showed additive effects, with heterozygous individuals having intermediate phenotypes (LUVp = 0.59 ± 0.18; *Figure 2d*). The association between floral UV patterns and the Chr15_LUVp SNP was confirmed in the F$_2$ progeny of crosses between plants homozygous for the L allele (with completely UV-absorbing ligules; LUVp = 1) and for the S allele (with a small UV-absorbing patch at the ligule base; LUVp < 0.18; *Figure 2e*, *Figure 2—figure supplement 2*). Average LUVp values were lower, and their range narrower, when these populations were grown in a greenhouse rather than in a field. Plants in the greenhouse experienced relatively uniform temperatures and humidity, and were shielded from most UV radiation. These results suggest that although floral UV patterns have a strong genetic basis (consistent with previous observations; *Koski and Ashman, 2013*), their expression is also affected by the environment.

## *HaMYB111* regulates UV pigment production

While no obvious candidate genes were found for the GWAS peaks for floral UV pigmentation in *H. petiolaris petiolaris*, the *H. annuus* chromosome 15 peak is ~5 kbp upstream of *HaMYB111*, a sunflower homolog of the *Arabidopsis thaliana AtMYB111* gene (*Figure 2b*). Together with AtMYB11 and AtMYB12, AtMYB111 is part of a small family of transcription factors (also called PRODUCTION OF FLAVONOL GLYCOSIDES [PFG]) that controls the expression of genes involved in the production of flavonol glycosides in *Arabidopsis* (*Stracke et al., 2007*). Flavonol glycosides are a subgroup of flavonoids known to fulfil a variety of functions in plants, including protection against abiotic and biotic stresses (e.g., UV radiation, cold, drought, herbivory) (*Pollastri and Tattini, 2011*). Crucially, they absorb strongly in the near UV range (300–400 nm) and are the pigments responsible for floral UV patterns in several plant species (*Rieseberg and Schilling, 1985*; *Sheehan et al., 2016*; *Thompson et al., 1972*). For instance, alleles of a homolog of *AtMYB111* are responsible for the evolutionary gain and subsequent loss of flavonol accumulation and UV absorption in flowers of *Petunia* species,

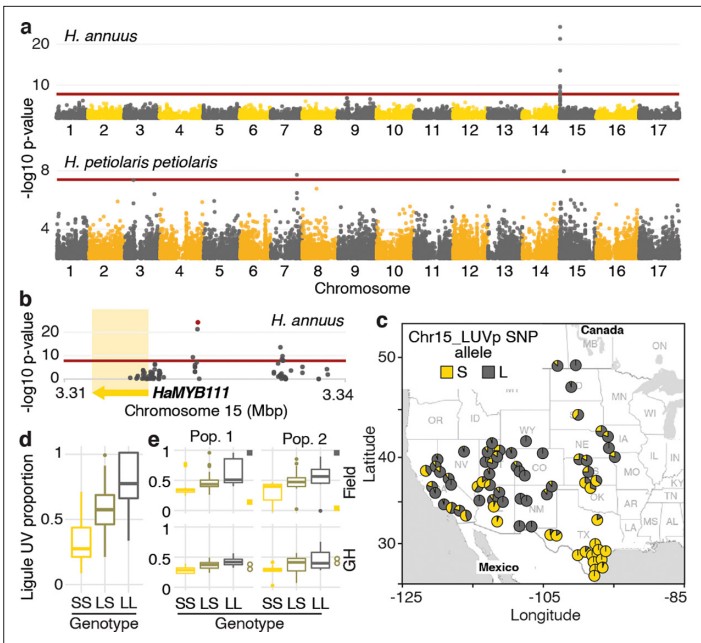

**Figure 2.** A single locus explains most of the variation in floral ultraviolet (UV) patterning in *H. annuus*. (**a**) Ligule ultraviolet proportion (LUVp) genome-wide association studies (GWAS). (**b**) Zoomed-in Manhattan plot for the chromosome 15 LUVp peak in *H. annuus*. Red lines represent 5% Bonferroni-corrected significance. GWAS were calculated using two-sided mixed models. Number of individuals: n = 563 individuals (*H. annuus*); n = 159 individuals (*H. petiolaris petiolaris*). Only positions with -log10 p-value >2 are plotted. *HaMYB111* is the only annotated feature in the genomic interval shown in *Figure 1b*; the single-nucleotide polymorphism (SNP) with the strongest association to LUVp (Chr15_LUVp SNP) is highlighted in red. Linkage disequilibrium (LD) decays rapidly in wild *H. annuus* (average $R^2$ at 10 kbp is ~0.035; *Todesco et al., 2020*), and all SNPs significantly associated with LUVp in *H. annuus* are included in the depicted region. The chromosome 15 association in *H. petiolaris petiolaris* is distinct from the one in *H. annuus* as it is located ~20 Mbp downstream of *HaMYB111*. (**c**) Geographical distribution of Chr15_LUVp SNP allele frequencies in *H. annuus*. L, large; S, small allele. (**d**) LUVp associated with different genotypes at Chr15_LUVp SNP in natural populations of *H. annuus* grown in a common garden. All pairwise comparisons are significant for $p<10^{-16}$ (one-way ANOVA with post-hoc Tukey HSD test, *F* = 438, df = 2; n = 563 individuals). LUVp values for the individuals in the GWAS populations and genotype data for Chr15_LUVp SNP are reported in *Figure 1—source data 2*. (**e**) LUVp associated with different genotypes at Chr15_LUVp SNP in *H. annuus* $F_2$ populations grown in the field or in a greenhouse (GH). Measurements for the parental generations are shown: squares, grandparents (field-grown); empty circles, $F_1$ parents (GH-grown; *Figure 2—figure supplement 2*). Boxplots show the median, box edges represent the 25th and 75th percentiles, whiskers represent the maximum/minimum data points within 1.5× interquartile range outside box edges. Differences between genotypic groups are significant for p=0.0057 (Pop. 1 Field, one-way ANOVA, *F* = 5.73, df = 2; n = 54 individuals); p=0.0021 (Pop. 2 Field, one-way ANOVA, *F* = 7.02, df = 2; n = 50 individuals); p=0.00015 (Pop. 1 GH, one-way ANOVA, *F* = 11.13, df = 2; n = 42 individuals); p=0.054 (Pop. 2 GH, one-way ANOVA, *F* = 3.17, df = 2; n = 38 individuals). p-Values for pairwise comparisons for panels (**d**) and (**e**) are reported in *Figure 2—source data 1*.

The online version of this article includes the following source data and figure supplement(s) for figure 2:

**Source data 1.** Ligule ultraviolet proportion (LUVp) values and Chr15_LUVp SNP genotypes for $F_2$.

**Figure supplement 1.** Ligule ultraviolet proportion (LUVp) genome-wide association study (GWAS) in *H. petiolaris fallax* (n = 193 individuals).

**Figure supplement 2.** Floral ultraviolet (UV) patterns in the parental lines of $F_2$ populations.

**Figure supplement 3.** Ligule ultraviolet proportion (LUVp) genome-wide association study (GWAS) in unfiltered *H. annuus* datasets.

associated with two successive switches in pollinator preferences (from bees, to hawkmoths, to hummingbirds; *Sheehan et al., 2016*). A homolog of *AtMYB12* has also been associated with variation in floral UV patterns in *Brassica rapa* (*Brock et al., 2016*). Analysis of sunflower ligules found two main groups of UV-absorbing compounds: glycoside conjugates of quercetin (a flavonol) and di-*O*-caffeoyl

quinic acid (CQA, a member of a family of antioxidant compounds that includes chlorogenic acid and that accumulates at high levels in many sunflower tissues; *Koeppe et al., 1970*). Both quercetin glycosides and CQA were more abundant at the base of sunflower ligules, and in ligules of plants with larger LUVp. However, this pattern was much more dramatic for flavonols, and they represented a much larger fraction of the total UV absorbance in UV-absorbing (parts of) ligules, suggesting that flavonols are the main pigments responsible for UV patterning in sunflower ligules (*Figure 3a and b*).

In *Arabidopsis*, *AtMYB12* and *AtMYB111* are known to have the strongest effect on flavonol glyco- side accumulation (*Stracke et al., 2007*; *Stracke et al., 2010*). We noticed, from existing RNAseq data, that *AtMYB111* expression levels are particularly high in petals (*Klepikova et al., 2016*; *Figure 3c*) and found that *Arabidopsis* petals, while uniformly white in the visible spectrum, absorb strongly in the UV (*Figure 3d*). To our knowledge, this is the first report of floral UV pigmentation in *Arabidopsis*, a highly selfing species that is seldom insect-pollinated (*Hoffmann et al., 2003*). Accumulation of flavonol glycosides is strongly reduced, and UV pigmentation is almost completely absent, in petals of mutants for *AtMYB111* (*myb111*). UV absorbance is further reduced in petals of double mutants for *AtMYB12* and *AtMYB111* (*myb12/111*). However, petals of the single mutant for *AtMYB12* (*myb12*), which is expressed at low levels throughout the plant (*Klepikova et al., 2016*), are indistinguishable from wild-type plants (*Figure 3d and e*). This shows that flavonol glycosides are responsible for floral UV pigmentation also in *Arabidopsis*, and that *AtMYB111* plays a fundamental role in controlling their accumulation in petals.

To confirm that sunflower *HaMYB111* is functionally equivalent to its *Arabidopsis* homolog, we introduced it into *myb111* plants. Expression of *HaMYB111*, either under the control of a constitutive promoter or of the endogenous *AtMYB111* promoter, restored petal UV pigmentation and induced accumulation of flavonol glycosides (*Figure 3d and e*). *HaMYB111* coding sequences obtained from wild sunflowers with large or small LUVp were equally effective at complementing the *myb111* mutant. Together with the observation that the strongest GWAS association with LUVp fell in the promoter region of *HaMYB111*, these results suggest that differences in the effect of the 'small' and 'large' alleles of this gene on floral UV pigmentation are not due to differences in protein function, but rather to differences in gene expression.

Analysis of *HaMYB111* expression patterns in cultivated sunflower revealed that, consistent with a role in floral UV pigmentation and similar to its *Arabidopsis* counterpart, it is expressed specifically in ligules, and it is almost undetectable in other tissues (*Badouin et al., 2017*; *Figure 3f*). Similar to observations in *Rudbeckia hirta*, another member of the *Heliantheae* tribe (*Schlangen et al., 2009*), UV pigmentation is established early in ligule development in both *H. annuus* and *H. petiolaris* as their visible colour turns from green to yellow before the inflorescence opens (R4 developmental stage; *Schneiter and Miller, 1981*; *Figure 3g*, *Figure 3—figure supplement 1*). *HaMYB111* is highly expressed in the part of the ligule that accumulates UV-absorbing pigments, and especially in developing ligules, consistent with a role in establishing pigmentation patterns (*Figure 3h*). We also observed a matching expression pattern for *HaFLS1*, the sunflower homolog of a gene encoding one of the main enzymes controlling flavonol biosynthesis in *Arabidopsis* (*FLAVONOL SYNTHASE 1*, *AtFLS1*), whose expression is regulated directly by *AtMYB111* (*Stracke et al., 2007*; *Figure 3i*). Finally, we compared *HaMYB111* expression levels in a set of 46 field-grown individuals with contrasting LUVp values, representing 21 different wild populations. *HaMYB111* expression levels differed significantly between the two groups (p=0.009; *Figure 3j*). Variation in expression levels within phenotypic classes was quite large; this is likely due at least in part to the strong dependence of *HaMYB111* expression on developmental stage (*Figure 3g*) and the difficulty of accurately establishing matching ligule devel- opmental stages across diverse wild sunflowers.

These expression analyses further point to *cis*-regulatory rather than coding sequence differences between *HaMYB111* alleles being responsible for LUVp variation. Accordingly, direct sequencing of the *HaMYB111* locus from multiple wild *H. annuus* individuals, using a combination of Sanger sequencing and long PacBio HiFi reads, identified no coding sequence variants associated with differ- ences in floral UV patterns, or with alleles at the Chr15_LUVp SNP (*Figure 3—figure supplement 2*, *Supplementary files 1 and 2*). However, we observed extensive variation in the promoter region of *HaMYB111*, differentiating wild *H. annuus* alleles from each other and from the reference assembly for cultivated sunflower (*Supplementary files 3 and 4*). Relaxing quality filters to include less well- supported SNPs in our LUVp GWAS did not identify additional variants with stronger associations

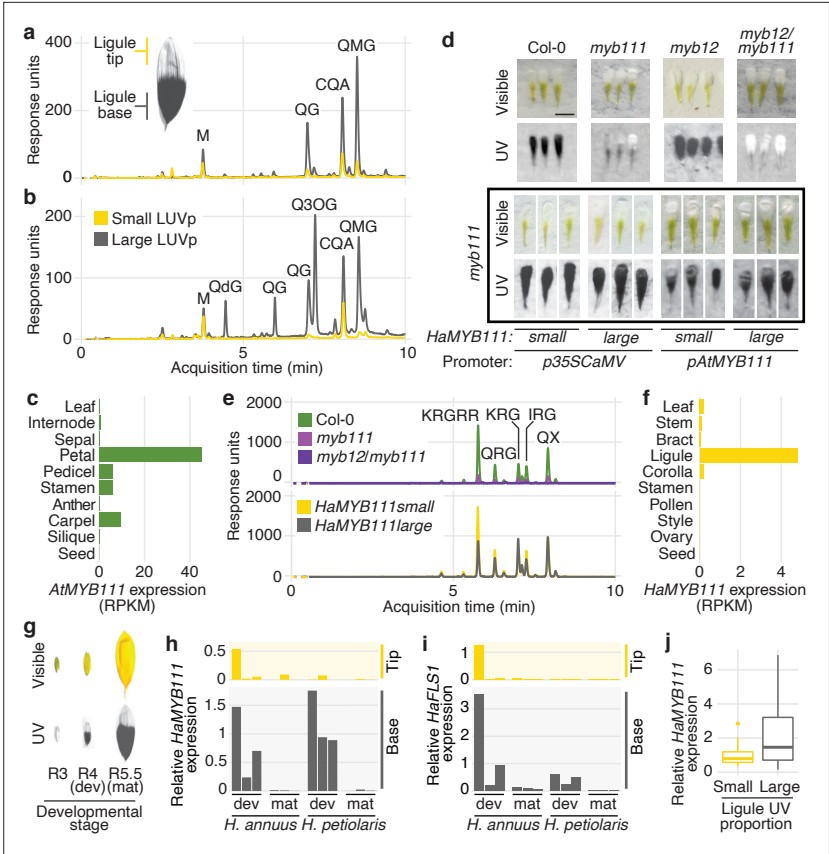

**Figure 3.** *MYB111* is associated with floral ultraviolet (UV) pigmentation patterns and flavonol accumulation in sunflower and *Arabidopsis*. (**a**) UV chromatograms (350 nm) for methanolic extracts of the upper and lower third of ligules with intermediate UV patterns, and (**b**) of ligules with large and small floral UV patterns. Peak areas are proportional to the total amount of absorbance at 350 nm explained by the corresponding compounds in the extracts. Relevant peaks are labelled: M, myricetin; QG, quercetin glucoside; QdG, quercetin diglucoside; Q3OG, quercetin-3-*O*-glucoside (co-elutes with quercetin-glucoronide); QMG, quercetin malonyl-glucoside; CQA, di-*O*-caffeoyl quinic acid (*Figure 3—source data 1*). (**c**) Expression levels of *AtMYB111* in *Arabidopsis*. RNAseq data were obtained from *Klepikova et al., 2016*. RPKM, reads per kilobase of transcript per million mapped reads. (**d**) *Arabidopsis* petals. *HaMYB111* from *H. annuus* plants with small or large ligule ultraviolet proportion (LUVp) was introduced into the *Arabidopsis myb111* mutant under the control of a constitutive promoter (*p35ScaMV*) or of the promoter of the *Arabidopsis* homolog (*pAtMYB111*). Scale bar = 1 mm. (**e**) UV chromatograms (350 nm) for methanolic extracts of petals of *Arabidopsis* lines. Upper panel: wild-type Col-0 and mutants. Bottom panel: *p35ScaMV::HaMYB111* lines in *myb111* background. Relevant peaks are labelled: KRGRR, kaempferol-rhamnoside-glucoside-rhamnoside-rhamnoside; QGR, quercetin-rhamnoside-glucoside; KRG, kaempferol-rhamnoside-glucoside; IRG, isorhamnetin-rhamnoside-glucoside; QX, quercetin-xyloside. (*Figure 3—source data 1*). (**f**) Expression levels of *HaMYB111* in the XRQ line of cultivated sunflower. RNAseq data were obtained from *Badouin et al., 2017*. (**g**) Pigmentation patterns in ligules of wild *H. annuus* at different developmental stages: R3, closed inflorescence bud; R4, inflorescence bud opening; R5, inflorescence fully opened. (**h**) Expression levels in the UV-absorbing base (grey) and UV-reflecting tip (yellow) of mature (mat; collected from inflorescences at R5 stage) and developing (dev; collected from inflorescences at R4 stage) ligules for *HaMYB111* and (**i**) *HaFLS1*, one of its putative targets. One representative individual with intermediate LUVp values was chosen for each species. Each bar represents average expression over three technical replicates for a biological replicate (different inflorescence from the same individual). For each gene, expression data are normalized to the average expression levels in the base of developing ligules of *H. annuus*. (**j**) *HaMYB111* expression levels in ligules of field-grown wild *H. annuus* with contrasting floral UV pigmentation patterns. Expression data are normalized to the average expression levels across all the samples. The difference between the two groups is significant for p=0.009 (Welch *t*-test, *t* = 2.81, df = 27.32, two-sided; n = 24 individuals for the large LUVp group; n = 22 individuals for the small LUVp group). Similar correlations are observed when *HaMYB111* expression levels are compared to individuals' LUVp values or their genotype at the Chr15_LUVp SNP (see *Figure 3—source data 2*). Boxplots show the median, box edges represent the 25th and 75th percentiles, whiskers represent the maximum/minimum data points within

*Figure 3 continued on next page*

*Figure 3 continued*

1.5× interquartile range outside box edges.

The online version of this article includes the following source data and figure supplement(s) for figure 3:

**Source data 1.** Flavonols in methanolic extractions of sunflower ligules and *Arabidopsis* petals.

**Source data 2.** Expression analyses in sunflower and *Arabidopsis*.

**Figure supplement 1.** Stages of ligule development in *H. petiolaris*.

**Figure supplement 2.** Coding sequence alignment for *HaMYB111*.

than Chr15_LUVp SNP (*Figure 2—figure supplement 2*). However, many of the polymorphisms we identified by direct sequencing were either larger insertions/deletions (indels) or fell in regions that were too repetitive to allow accurate mapping of short reads, and would not be included even in this expanded SNP dataset. While several of these variants in the promoter region of *HaMYB111* appeared to be associated with the Chr15_LUVp SNP, further studies will be required to confirm this, and to identify their eventual effects on *HaMYB111* activity (see discussion in the legend of *Figure 3— figure supplement 2*).

Interestingly, when we sequenced the promoter region of *HaMYB111* in several *H. argophyllus* and *H. petiolaris* individuals, we found that they all carried the S allele at the Chr15_LUVp SNP, and that their promoter regions were generally more similar in sequence to those of *H. annuus* individuals carrying the S allele at the Chr15_LUVp SNP (*Supplementary files 3 and 4*). Similarly, in a set of previously re-sequenced wild sunflowers, we found the S allele to be fixed in several perennial (*Helianthus decapetalus*, *Helianthus divaricatus*, and *Helianthus grosseserratus*) and annual sunflower species (*H. argophyllus*, *Helianthus niveus*, *Helianthus debilis*), and to be at >0.98 frequency in *H. petiolaris* (*Figure 1—source data 2*). Conversely, the L allele at Chr15_LUVp SNP was almost fixed (>0.98 frequency) in a set of 285 cultivated sunflower lines (*Mandel et al., 2013*). Consistent with these patterns, UV bullseyes are considerably smaller in *H. argophyllus* (mean LUVp ± SD = 0.27 ± 0.09), *H. niveus* (0.15 ± 0.09), and *H. petiolaris* (0.27 ± 0.12; *Figure 1e*) than in cultivated sunflower lines (0.62 ± 0.23). Additionally, while 50 of the cultivated sunflower lines had completely or almost completely UV-absorbing ligules (LUVp > 0.8), no such case was observed in the other three species (*Figure 1—figure supplement 3*).

## A dual role for floral UV pigmentation

Although our results show that *HaMYB111* explains most of the variation in floral UV pigmentation patterns in wild *H. annuus*, why such variation exists in the first place is less clear. Several hypotheses have been advanced to explain the presence of floral UV patterns and their variability. Like their visible counterparts, UV pigments play a fundamental role in pollinator attraction (*Horth et al., 2014*; *Koski et al., 2014*; *Rae and Vamosi, 2013*; *Sheehan et al., 2016*). For example, in *Rudbeckia* species, artificially increasing the size of bullseye patterns to up to 90% of the petal surface resulted in rates of pollinator visitation equal to or higher than wild-type flowers (which have on average 40–60% of the petal being UV-absorbing). Conversely, reducing the size of the UV bullseye had a strong negative effect on pollinator visitation (*Horth et al., 2014*). To test whether the relative size of UV bullseye patterns affected pollination, we assessed insect visitation rates for wild *H. annuus* lines with contrasting UV bullseye patterns. An initial experiment compared inflorescences from pairs of plants from two populations (ANN_03 from California and ANN_55 from Texas), which were selected to have large or small floral UV patterns. In this setup, inflorescences with large UV patterns received significantly more visits (*Figure 4a*). While this experiment revealed a clear pattern of pollinator preferences, it involved plants from only two different populations, and effects of other unmeasured factors unrelated to UV pigmentation on visitation patterns cannot be excluded. Therefore, we monitored pollinator visitation in plants grown in a common garden experiment including 1484 individuals from 106 *H. annuus* populations, spanning the entire range of the species. Assaying a much more diverse population of *H. annuus* individuals should reduce effects on pollinator preferences of traits unrelated to floral UV pigmentation. Within this field, we selected 82 plants, from 49 populations, which flowered at roughly the same time and had comparable numbers of flowers. We selected plants falling into three categories of LUVp values, representatives of the more abundant phenotypic classes across the range of wild *H. annuus* (*Figure 1d*): small (LUVp = 0–0.3), intermediate (LUVp = 0.5–0.8), and large (LUVp >0.95).

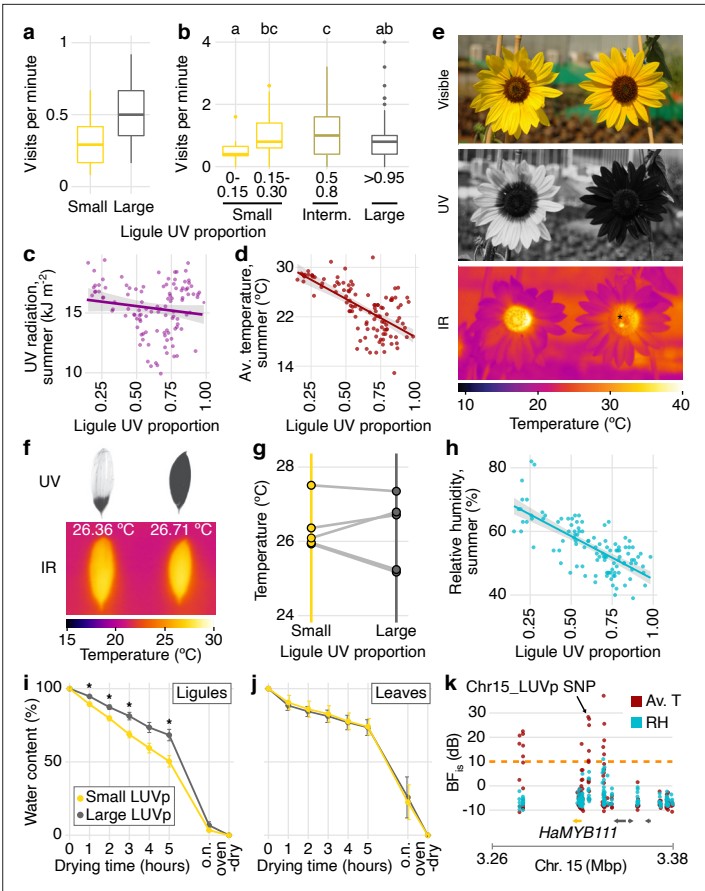

**Figure 4.** Accumulation of ultraviolet (UV) pigments in flowers affects pollinator visits and transpiration rates. (**a**) Rates of pollinator visitation measured in Vancouver in 2017 (p=0.017; Mann–Whitney $U$-tests, W = 150, two-sided; n = 143 pollinator visits) and (**b**) 2019 (differences between ligule ultraviolet proportion [LUVp] categories are significant for p=0.0058, Kruskal–Wallis test, $\chi^2$ = 14.54, df = 4; n = 1390 pollinator visits). Letters identify groups that are significantly different for p<0.05 in pairwise comparisons, Wilcoxon rank sum test. Exact p-values are reported in *Figure 4—source data 1–4*. Boxplots show the median, box edges represent the 25th and 75th percentiles, whiskers represent the maximum/minimum data points within 1.5× interquartile range outside box edges. (**c**) Correlation between average LUVp for different populations of *H. annuus* and summer UV radiation ($R^2$ = 0.01, p=0.12, n = 110 populations) or (**d**) summer average temperature ($R^2$ = 0.44, p=2.4 × $10^{-15}$, n = 110 populations). Grey areas represent 95% confidence intervals. (**e**) Sunflower inflorescences pictured in the visible, UV, and infrared (IR) range. In the IR picture, a bumblebee is visible in the inflorescence with large LUVp (right; the warmer abdomen of the bee is visible as a bright yellow spot under the asterisk). The higher temperature in the centre (disc) of the inflorescence with small LUVp does not depend on ligule UV patterns (*Figure 4—figure supplement 3*). These inflorescences belong to two of the plants that were used for the pollination preference experiments reported in (**a**) and are representative of the differences in floral UV patterns between LUVp categories in that experiment. (**f**) *H. annuus* ligules after having been exposed to sunlight for 15 min. (**g**) Five pairs of ligules from different sunflower lines were exposed to sunlight for 15 min, and their average temperature was measured from IR pictures. (**h**) Correlation between average LUVp in *H. annuus* populations and summer relative humidity (RH) ($R^2$ = 0.51, p=1.4 × $10^{-18}$, n = 110 populations). The grey area represents the 95% confidence interval. (**i**) Rate of water loss from ligules and (**j**) leaves of wild *H. annuus* plants with large or small LUVp. Values reported are means ± standard error of the mean. n = 16 inflorescences (ligules) or 15 plants (leaves). Three detached ligules and one or two leaves for each individual were left to air-dry and weighed every hour for 5 hr, after they were left to air-dry overnight (o.n.), and after they were incubated in an oven to remove any residual humidity (oven-dry). Asterisks denote significant differences (p<0.05, two-sided Welch $t$-test; exact p-values are reported in *Figure 4—source data 1–4*). (**k**) Genotype-environment association (GEA) for summer average temperature (Av. T) and summer RH in the *HaMYB111* region. The dashed orange line represents Bayes factor ($BF_{is}$) = 10 deciban (dB). GEAs were calculated using two-sided XtX statistics. n = 71 populations.

The online version of this article includes the following source data and figure supplement(s) for figure 4:

*Figure 4 continued on next page*

*Figure 4 continued*

**Source data 1.** Pollinator experiment data.

**Source data 2.** Temperature measurements from infrared pictures for individual detached ligules.

**Source data 3.** Ligules and leaves desiccation experiment data.

**Source data 4.** Genotype-environment association (GEA) results for the *HaMYB111* region.

**Figure supplement 1.** Pollinator visits in the 2019 field experiment divided by category of pollinators.

**Figure supplement 2.** Correlations between ligule ultraviolet proportion (LUVp) and environmental variables in *H. annuus*.

**Figure supplement 3.** Inflorescence temperature time series.

**Figure supplement 4.** Correlations between ligule ultraviolet proportion (LUVp) and other floral characteristics.

**Figure supplement 5.** Correlations between ligule ultraviolet proportion (LUVp) and environmental variables in *H. petiolaris*.

---

Plants with intermediate UV patterns had the highest visitation rates (*Figure 4b*, *Figure 4—figure supplement 1*). Visitation to plants with small or large UV patterns was less frequent, and particularly low for plants with very small LUVp values (<0.15). Pollination rates are known to be yield-limiting in sunflower (*Greenleaf and Kremen, 2006*), and a strong reduction in pollination could therefore have a negative effect on fitness; this would be consistent with the observation that plants with very small LUVp values were rare (~1.5% of individuals) in our common garden experiment, which was designed to provide a balanced representation of the natural range of *H. annuus*. Although pollinator preferences in this experiment could still be affected by other unmeasured factors (nectar content, floral volatiles), these results are consistent with previous results showing that floral UV patterns play a major role in pollinator attraction (*Horth et al., 2014*; *Koski et al., 2014*; *Rae and Vamosi, 2013*; *Sheehan et al., 2016*). They also agree with earlier findings in other plant species, suggesting that intermediate-to-large UV bullseyes are preferred by pollinators (*Horth et al., 2014*; *Koski et al., 2014*). While we cannot exclude that smaller UV bullseyes would be preferred by pollinators in some parts of the *H. annuus* range, this does not seem likely; the most common pollinators of sunflower are ubiquitous across the range of *H. annuus*, and many bee species known to pollinate sunflower are found in both regions where *H. annuus* populations have large LUVp and regions where they have small LUVp (*Hurd et al., 1980*). Therefore, while acting as visual cues for pollinators is clearly a major function of floral UV bullseyes, it is unlikely to (fully) explain the patterns of variation that we observe for this trait.

In recent years, the importance of non-pollinator factors in driving selection for floral traits has been increasingly recognized (*Strauss and Whittall, 2006*). Additionally, flavonol glycosides, the pigments responsible for floral UV patterns in sunflower, are known to be involved in responses to several abiotic stressors (*Korn et al., 2008*; *Nakabayashi et al., 2014b*; *Pollastri and Tattini, 2011*; *Schulz et al., 2015*). Therefore, we explored whether some of these stressors could drive diversification in floral UV pigmentation. An intuitively strong candidate is UV radiation, which can be harmful to plant cells (*Stapleton, 1992*). Variation in the size of UV bullseye patterns across the range of *Argentina anserina* (a member of the *Rosaceae* family) has been shown to correlate positively with intensity of UV radiation. Flowers of this species are bowl-shaped, and larger UV-absorbing regions have been proposed to protect pollen from UV damage by absorbing UV radiation that would otherwise be reflected toward the anthers (*Koski and Ashman, 2015*). However, sunflower inflorescences are much flatter than *A. anserina* flowers, making it unlikely that any significant amount of UV radiation would be reflected from the ligules towards the disc flowers. Studies in another plant with non-bowl-shaped flowers (*Clarkia unguiculata*) have found no evidence of an effect of floral UV patterns in protecting pollen from UV damage (*Peach et al., 2020*). Consistent with this, the associations between the intensity of UV radiation at our collection sites and floral UV patterns in *H. annuus* was weak (*H. annuus: R²* = 0.01, p=0.12; *Figure 4c*, *Figure 4—figure supplement 2*).

Across the *Potentillae* tribe (*Rosaceae*), floral UV bullseye size is also weakly associated with UV radiation, but is more strongly correlated with temperature, with lower temperatures being associated with larger UV bullseyes (*Koski and Ashman, 2016*). We found a similar, strong correlation with temperature in our dataset, with lower average summer temperatures being associated with larger LUVp values in *H. annuus* ($R^2$ = 0.44, p=2.4 × 10⁻¹⁵; *Figure 4d*, *Figure 4—figure supplement 2*). It has

been suggested that the radiation absorbed by floral UV pigments could contribute to increasing the temperature of the flower, similar to what has been observed for visible pigments (*Koski et al., 2020*). This possibility is particularly intriguing for sunflower, in which flower temperature plays an important role in pollinator attraction; inflorescences of cultivated sunflowers consistently face east so that they warm up faster in the morning, making them more attractive to pollinators (*Atamian et al., 2016*; *Creux et al., 2021*). By absorbing more radiation, larger UV bullseyes could therefore contribute to increasing temperature of the sunflower inflorescences, and their attractiveness to pollinators, in cold climates. However, UV wavelengths represents only a small fraction (3–7%) of the solar radiation reaching the Earth's surface (compared to >50% for visible wavelengths), and might therefore not provide sufficient energy to significantly warm up the ligules (*Nunez et al., 1994*). In line with this observation, different levels of UV pigmentation had no effect on the temperature of inflorescences or individual ligules exposed to sunlight (*Figure 4e–g*, *Figure 4—figure supplement 3*).

While several geoclimatic variables are correlated across the range of wild *H. annuus*, the single variable explaining the largest proportion of the variation in floral UV patterns in this species was summer relative humidity (RH; $R^2 = 0.51$, p=$1.4 \times 10^{-18}$; *Figure 4h*, *Figure 4—figure supplement 2*), with lower humidity being associated with larger LUVp values (i.e., higher concentrations of flavonol glycosides in ligules). Lower RH is generally associated with higher transpiration rates in plants, leading to increased water loss, and flavonol glycosides are known to play an important role in responses to drought stress (*Nakabayashi et al., 2014a*); in particular, *Arabidopsis* lines that accumulate higher concentrations of flavonol glycosides due to overexpression of *AtMYB12* lose water and desiccate at slower rates than wild-type plants (*Nakabayashi et al., 2014b*). Similarly, in a set of plants representing seven independent natural populations of *H. annuus*, we found that completely UV-absorbing ligules desiccate at a significantly slower rate than largely UV-reflecting ligules (*Figure 4i*). This is not due to general differences in transpiration rates between genotypes since we observed no comparable trend for rates of leaf desiccation in the same set of sunflower lines (*Figure 4j*). Transpiration from flowers can be a major source of water loss for plants, and this is known to drive, within species, the evolution of smaller flowers in populations living in dry locations (*Galen, 2000*; *Herrera, 2005*; *Lambrecht, 2013*; *Lambrecht and Dawson, 2007*; see *Figure 4—figure supplement 4*). While desiccation rates are only a proxy for transpiration in field conditions (*Duursma et al., 2019*; *Hygen, 1951*), and other factors might affect ligule transpiration in this set of lines, this evidence (strong correlation between LUVp and summer RH; known role of flavonol glycosides in regulating transpiration; and correlation between extent of ligule UV pigmentation and desiccation rates) suggests that variation in floral UV pigmentation in sunflowers is driven by the role of flavonol glycosides in reducing water loss from ligules, with larger floral UV patterns helping prevent drought stress in drier environments.

One of the main roles of transpiration in plants is facilitating heat dispersion at higher temperatures through evaporative cooling (*Burke and Upchurch, 1989*; *Drake et al., 2018*), which could explain the strong correlation between LUVp and temperature across the range of *H. annuus* (*Figure 4d*). Consistent with this, summer RH and summer temperatures together explain a considerably larger fraction of the variation for LUVp in *H. annuus* than either variable alone ($R^2 = 0.63$, p=0.0017; *Figure 1—source data 1*), with smaller floral UV patterns being associated with higher RH and higher temperatures (*Figure 4—figure supplement 2*). Consistent with a role of floral UV pigmentation in the plant's response to variation in both humidity and temperature, we found strong associations (dB > 10) between SNPs in the *HaMYB111* region and these variables in genotype-environment association (GEA) analyses (*Figure 4k*, *Figure 4—source data 4*). Despite a more limited range of variation for LUVp, a similar trend (larger UV patterns in drier, colder environments) is present also in *H. petiolaris* (*Figure 4—figure supplement 5*). Interestingly, while the L allele at Chr_15 LUVp SNP is present in *H. petiolaris* (*Figure 1—figure supplement 3*), it is found only at a very low frequency and does not seem to significantly affect floral UV patterns in this species (*Figure 2a*). This could represent a recent introgression since *H. annuus* and *H. petiolaris* are known to hybridize in nature (*Heiser, 1947*; *Yatabe et al., 2007*). Alternatively, the Chr_15 LUVp SNP might not be associated with functional differences in *HaMYB111* in *H. petiolaris*, or differences in genetic networks or physiology between *H. annuus* and *H. petiolaris* could mask the effect of this allele, or limit its adaptive advantage, in the latter species.

## Conclusions

Connecting adaptive variation to its genetic basis is one of the main goals of evolutionary biology. Here, we show that regulatory variation at a single major gene, the transcription factor *HaMYB111*, underlies most of the diversity for floral UV patterns in the common sunflower, wild *H. annuus*. Variation for these floral UV patterns correlates strongly with pollinator preferences, but also with geoclimatic variables (especially RH and temperature) and desiccation rates in sunflower ligules. While the effects of floral UV patterns on pollinator attraction are well-known, these associations suggest a role of environmental factors in shaping diversity for this trait. Larger floral UV patterns, due to accumulation of flavonol glycoside pigments in ligules, could help reduce the amount of transpiration in environments with lower RH, preventing excessive water loss and maintaining ligule turgidity. In humid, hot environments (e.g., Southern Texas), lower accumulation of flavonol glycosides would instead promote transpiration from ligules, keeping them cool and avoiding overheating. The presence of UV pigmentation in the petals of *Arabidopsis* (also controlled by the *Arabidopsis* homolog of *MYB111*) further points to a more general protective role of these pigments in flowers since pollinator attraction is likely not critical for fertilization in this largely selfing species. It should be noted that, while we have examined some of the most likely factors explaining the distribution of variation for floral UV patterns in wild *H. annuus* across North America, other abiotic factors could play a role, as well as biotic ones (e.g., the aforementioned differences in pollinator assemblages, or a role of UV pigments in protection from herbivory; *Gronquist et al., 2001*). However, a role of floral UV patterns in reducing water loss from petals is consistent with the overall trend in increased size of floral UV patterns over the past 80 years that has been observed in herbarium specimens (*Koski et al., 2020*); due to changing climates, RH over land has been decreasing in recent decades, which could result in higher transpiration rates (*Byrne and O'Gorman, 2018*). Further studies will be required to confirm the existence of this trend and assess its strength.

More generally, our study highlights the complex nature of adaptive variation, with selection pressures from both biotic and abiotic factors shaping the patterns of diversity that we observe across natural populations. Floral diversity in particular has long been attributed to the actions of animal pollinators. Our work adds to a growing literature demonstrating the contributions of abiotic factors to this diversity.

## Materials and methods

**Key resources table**

| Reagent type (species) or resource | Designation | Source or reference | Identifiers | Additional information |
|---|---|---|---|---|
| Gene (*Helianthus annuus*) | HaMYB111 | INRA Sunflower Bioinformatics Resources | HanXRQChr15g0465131 | |
| Gene (*H. annuus*) | HaFLS1 | INRA Sunflower Bioinformatics Resources | HanXRQChr09g0258321 | |
| Gene (*H. annuus*) | HaEF1α | INRA Sunflower Bioinformatics Resources | HanXRQChr11g0334971 | |
| Gene (*Arabidopsis thaliana*) | AtMYB111; PFG3 | The Arabidopsis Information Resource | At5G49330 | |
| Gene (*A. thaliana*) | AtMYB12; PFG1 | The Arabidopsis Information Resource | At2G47460 | |
| Strain, strain background (*Helianthus* spp.) | Various *Helianthus* species and individuals | USDA, North Central Regional Plant Introduction Station | | See *Figure 1—source data 1* for full list |
| Strain, strain background (*A. thaliana*) | Col-0 | Arabidopsis Biological Resource Center | CS28167 | |

*Continued on next page*

*Continued*

| Reagent type (species) or resource | Designation | Source or reference | Identifiers | Additional information |
|---|---|---|---|---|
| Genetic reagent (*A. thaliana*) | *myb111* | Arabidopsis Biological Resource Center | CS9813 | |
| Genetic reagent (*A. thaliana*) | *myb12* | Arabidopsis Biological Resource Center | CS9602 | |
| Genetic reagent (*A. thaliana*) | *myb12/myb111* | Arabidopsis Biological Resource Center | CS9980 | |
| Recombinant DNA reagent | *p35SCaMV:: HaMYB111 large* | This paper | | *HaMYB111* CDS from sunflower lines with large LUVp, constitutive promoter |
| Recombinant DNA reagent | *p35SCaMV:: HaMYB111 small* | This paper | | *HaMYB111* CDS from sunflower lines with small LUVp, constitutive promoter |
| Recombinant DNA reagent | *pAtMYB111:: HaMYB111 large* | This paper | | *HaMYB111* CDS from sunflower lines with large LUVp, endogenous *Arabidopsis* promoter |
| Recombinant DNA reagent | *pAtMYB111:: HaMYB111 small* | This paper | | *HaMYB111* CDS from sunflower lines with small LUVp, endogenous *Arabidopsis* promoter |
| Sequence-based reagent | *HaMYB111* CDS F | This paper | PCR primer | ATGGGAAGGACCCCGTGTT |
| Sequence-based reagent | *HaMYB111* CDS R | This paper | PCR primer | TTAAGACTGAAACCATGCATCTACC |
| Sequence-based reagent | *AtMYB111* promoter F | This paper | PCR primer | CCTGTGCTTTAAGGCTCGAC |
| Sequence-based reagent | *AtMYB111* promoter R | This paper | PCR primer | TGCTTCTCGGTCTCTTCTGT |
| Sequence-based reagent | *HaMYB111* qPCR F | This paper | PCR primer | ATGGGAAGGACCCCGTGTT |
| Sequence-based reagent | *HaMYB111* qPCR R | This paper | PCR primer | GCAACTCTTTCCGCATCTCA |
| Sequence-based reagent | *HaFLS1* qPCR F | This paper | PCR primer | AAACTACTACCCACCATGCC |
| Sequence-based reagent | *HaFLS1* qPCR R | This paper | PCR primer | TCCTTGTTCACTGTTGTTCTGT |
| Sequence-based reagent | *EF1α* qPCR F | This paper | PCR primer | GTGTGTGATGTCGTTCTCCA |
| Sequence-based reagent | *EF1α* qPCR R | This paper | PCR primer | ATTCCACCCAAAGCTTGCTC |
| Commercial assay or kit | CloneJET PCR cloning kit | Thermo Fisher Scientific | Cat. #: K1231 | |
| Commercial assay or kit | Custom TaqMan SNP Genotyping Assay | Thermo Fisher Scientific | Assay ID: ANKCD29 | |
| Commercial assay or kit | TaqMan Genotyping Master Mix | Thermo Fisher Scientific | Cat. #: 4371355 | |
| Commercial assay or kit | RevertAid RT Reverse Transcription Kit | Thermo Fisher Scientific | Cat. #: K1691 | |

*Continued on next page*

*Continued*

| Reagent type (species) or resource | Designation | Source or reference | Identifiers | Additional information |
|---|---|---|---|---|
| Commercial assay or kit | SsoFast EvaGreen Supermix | Bio-Rad | Cat. #: 1725201 | |
| Peptide, recombinant protein | Phusion High-Fidelity DNA Polymerase | Thermo Fisher Scientific | Cat. #: F530L | |
| Chemical compound, drug | PPM (Plant Preservative Mixture) | Plant Cell Technologies | | |
| Chemical compound, drug | TRIzol Reagent | Thermo Fisher Scientific | Cat. #: 15596026 | |
| Software, algorithm | ImageJ | ImageJ (https://imagej.nih.gov/ij/) | RRID:SCR_003070 | v2.0.0-rc-43/1.51o |
| Software, algorithm | Trimmomatic | Usadel lab (http://www.usadellab.org/cms/?page=trimmomatic) | RRID:SCR_011848 | v0.36 |
| Software, algorithm | NextGenMap | NextGenMap (https://cibiv.github.io/NextGenMap/) | RRID:SCR_005488 | v0.5.3 |
| Software, algorithm | GATK | Broad Institute (https://gatk.broadinstitute.org/hc/en-us) | RRID:SCR_001876 | v4.0.1.2 |
| Software, algorithm | Beagle | University of Washington (https://faculty.washington.edu/browning/beagle/beagle.html) | RRID:SCR_001789 | 10 Jun 18.811 |
| Software, algorithm | BWA | BWA (https://github.com/lh3/bwa) | RRID:SCR_010910 | v0.7.17 |
| Software, algorithm | EMMAX | University of Michigan – Center for Statistical Genetics (http://csg.sph.umich.edu//kang/emmax/download/index.html) | | v07 Mar 2010 |
| Software, algorithm | GEMMA | GEMMA (https://github.com/genetics-statistics/GEMMA) | | v0.98.3 |
| Software, algorithm | GCTA_GREML | GCTA (https://cnsgenomics.com/software/gcta) | | v1.93.2beta |
| Software, algorithm | BOLT-REML | BOLT-LMM (https://alkesgroup.broadinstitute.org/BOLT-LMM/BOLT-LMM_manual.html) | | v.2.3.5 |
| Software, algorithm | Agilent MassHunter | Agilent (https://www.agilent.com/en/promotions/masshunter-mass-spec) | RRID:SCR_015040 | |
| Software, algorithm | R | The R Project for Statistical Computing (https://www.r-project.org/) | RRID:SCR_001905 | v3.6.2 |

*Continued on next page*

*Continued*

| Reagent type (species) or resource | Designation | Source or reference | Identifiers | Additional information |
|---|---|---|---|---|
| Software, algorithm | 'raster' package | Spatial Data Science (https://rspatial.org/raster/index.html) | | |
| Software, algorithm | 'interactions' package | CRAN (https://cran.r-project.org/web/packages/interactions/) | | v1.1.5 |
| Software, algorithm | BayPass | BayPass (http://www1.montpellier.inra.fr/CBGP/software/baypass/) | | v2.1 |
| Software, algorithm | Fluke Connect | Fluke (https://www.fluke.com/en-ca/products/fluke-software/connect) | | v.1.1.536.0 |

## Plant material and growth conditions

Sunflower lines used in this paper were grown from seeds collected from wild populations (**Todesco et al., 2020**) or obtained from the North Central Regional Plant Introduction Station in Ames, IA. For all experiments except the plants used for **Figure 1—figure supplement 1b**, sunflower seeds were surface sterilized by immersion for 10 min in a 1.5% sodium hypochlorite solution. Seeds were then rinsed twice in distilled water and treated for at least 1 hr in a solution of 1% PPM (Plant Preservative Mixture; Plant Cell Technologies, Washington, DC), a broad-spectrum biocide/fungicide, to minimize contamination, and 0.05 mM gibberellic acid (Sigma-Aldrich, St. Louis, MO). They were then scarified, dehulled, and kept for 2 weeks at 4°C in the dark on filter paper moistened with a 1% PPM solution. Following this, seeds were kept in the dark at room temperature until they germinated. For common garden experiments, the seedlings were then transplanted in peat pots, grown in a greenhouse for 2 weeks, then moved to an open-sided greenhouse for a week for acclimation, and finally transplanted in the field at the Totem Plant Science Field Station of the University of British Columbia (Vancouver, Canada). For all other experiments, seedlings were transplanted in 2-gallon pots filled with Sunshine #1 growing mix (Sun Gro Horticulture Canada, Abbotsford, BC, Canada). Plants grown in greenhouses at the Vancouver campus of the University of British Columbia were kept at 26°C during the day and 20°C during the night, supplemented with LED light on a cycle of 16 hr days and 8 hr nights. For the wild sunflower species shown in **Figure 1—figure supplement 1b**, following sterilization, seeds were scarified and then dipped in fusicoccin solution (1.45 µM) for 15 min, dehulled, germinated in the dark for at least 8–10 days, and then grown in pots for 3 weeks before transplantation. One group of species was transplanted into 2-gallon pots filled with a blend of sandy loam, organic compost and mulch, and grown at the UC Davis Plant Sciences Field Station (Davis, CA) from July to October 2017. Several additional species were grown in single rows covered with mulch and spaced 0.75 m apart at the Oxford Tract Facility field (Berkeley, CA) from June to October 2021, or in a greenhouse facility at Berkeley, CA. A complete list of sunflower accessions and their populations of origin is reported in **Figure 1—source data 1** and **Figure 1—source data 2**.

Seeds from the following *Arabidopsis* lines were obtained from the Arabidopsis Biological Resource Center: Col-0 (CS28167), *myb111* (CS9813), *myb12* (CS9602), and *myb12/myb111* (CS9980). Seeds were stratified in 0.1% agar at 4°C in the dark for 4 days, and then sown in pots containing Sunshine #1 growing mix. Plants were grown in growth chambers at 23°C in long-day conditions (16 hr light, 8 hr dark).

## Common garden

Two common garden experiments were performed, in 2016 and 2019. After germination and acclimation, plants were transplanted at the Totem Plant Science Field Station of the University of British Columbia (Vancouver, Canada). In the 2016 common garden experiment, each sunflower species was grown in a separate field. Pairs of plants from the same population were randomly distributed within

each field. In the 2019 common garden experiment, plants were sown using a completely randomized design.

In the summer of 2016, 10 plants from each of the 151 selected populations of wild *H. annuus*, *H. petiolaris*, *H. argophyllus,* and *H. niveus* were grown. Plants were transplanted in the field on 25 May (*H. argophyllus*), 2 June (*H. petiolaris* and *H. niveus*), and 7 June 2016 (*H. annuus*). Up to four inflorescences from each plant were collected for visible and UV photography.

In the summer of 2019, 14 plants from each of the 106 populations of wild *H. annuus* were transplanted in the field on 6 June. These included 65 of the populations grown in the previous common garden experiment, and 41 additional populations that were selected to complement their geographical distribution. At least three ligules from at least two different inflorescences for each plant were collected for visible and UV photography. Ligules were selected to be as far apart from each other as possible across the inflorescence, taking care to avoid damaged or otherwise unrepresentative ligules.

Sample size for the common garden experiments was determined by the available growing space and resources. 10–14 individuals were grown for each population because this would provide a good representation of the variation present in each population, while maximizing the number of populations that could be surveyed. Researchers were not blinded as to the identity of individual samples. However, information about their populations of origin and/or LUVp phenotypes was not attached to the samples during data acquisition.

## Ultraviolet and infrared photography

Ultraviolet patterns were imaged in whole inflorescences or detached ligules (see 'Common garden' section) using a Nikon D70s digital camera, fitted with a Noflexar 35 mm lens and a reverse-mounted 2-inch Baader U-Filter (Baader Planetarium, Mammendorf, Germany), which only allows the transmission of light between 320 and 380 nm. Wild sunflower species shown in *Figure 1—figure supplement 1b* were imaged using a Canon DSLR camera in which the internal hot mirror filter had been replaced with a UV bandpass filter (LifePixel, Mukilteo, WA). Floral UV patterns were scored as LUVp, rather than total area or diameter of the UV bullseye, because LUVp is less influenced by genetic or environmental factors affecting inflorescence size (*Moyers et al., 2017*). The length of the whole ligule ($L_L$) and the length of the UV-absorbing part at the base of the ligule ($L_{UV-abs}$) were measured using ImageJ (*Schindelin et al., 2012*; *Schneider et al., 2012*). LUVp was measured as the ratio between the two ($LUVp = L_{UV-abs}/L_L$). In some *H. annuus* individuals, the upper, 'UV-reflecting' portion of the ligules ($L_{UV-ref}$) also displayed a lower level of UV absorption; in those cases, these regions were weighted at 50% of fully UV-absorbing regions using the formula $LUVp = (L_{UV-abs}/L_L) + ½(L_{UV-ref}/L_L)$. Partial UV absorbance in the tip of ligules was more common in plants with larger floral UV patterns (*Figure 1—figure supplement 2*). To avoid possible confounding effects, for all experiments plants in the 'small' and 'intermediate' LUVp classes were selected to have no noticeable UV absorbance in the tips of ligules. For UV pictures of whole inflorescences, LUVp values were measured for three representative ligules chosen to be as far apart from each other as possible, and the average of those three values was used as the LUVp for the inflorescence. LUVp values for all the inflorescences or detached ligules available for each plant were averaged to obtain the LUVp value for that individual.

Infrared pictures for the experiments shown in *Figure 4e–g* and *Figure 4—figure supplement 3* were taken using a Fluke TiX560 thermal imager (Fluke Corporation, Everett, WA) and analysed using the Fluke Connect software (v1.1.536.0). For time-series experiments on whole inflorescences, plants from populations ANN_03 (from CA, USA, with large LUVp) and ANN_55 (from TX, USA, with small LUVp) were germinated as above (see 'Common garden'), grown in 2-gallon pots in a greenhouse until they produced four true leaves, and then moved to the field. On three separate days in August 2017, pairs of inflorescences with opposite floral UV patterns at similar developmental stages were selected and made to face east. Infrared images were taken just before sunrise, ~5 min after sunrise, and then at 0.5, 1, 2, 3, and 4 hr after sunrise.

For infrared pictures of detached ligules, plants were grown in a greenhouse. Plants with large LUVp came from populations ANN_03 (CA, USA), ANN_16 (NM, USA), and ANN_19 (NM, USA); plants with small LUVp came from populations ANN_55 and ANN_58 (both from TX, USA). Flowerheads were collected and kept overnight in a room with constant temperature of 21°C, with their stems immersed in a beaker containing distilled water. The following day, pairs of inflorescences were randomly selected from the two LUVp categories, and representative, undamaged ligules were

removed and arranged on a sheet of white paper. Infrared pictures were taken immediately before exposing the ligules to the sun, and again 5, 10, and 15 min after that, at around 1 pm on 5 October 2020 (*Figure 4—source data 2*).

## Library preparation, sequencing, and SNP calling

Whole-genome shotgun (WGS) sequencing library preparation and sequencing, as well as SNP calling and variant filtering, for the *H. annuus* and *H. petiolaris* individuals used for GWAS analyses in this paper were previously described (*Todesco et al., 2020*). Briefly, DNA was extracted from leaf tissue using a modified CTAB protocol (*Murray and Thompson, 1980*; *Zeng et al., 2002*), the DNeasy Plant Mini Kit, or a DNeasy 96 Plant Kit (QIAGEN, Hilden, Germany). Genomic DNA was sheared to an average fragment size of 400 bp using a Covaris M220 ultrasonicator (Covaris, Woburn, MA). Libraries were prepared using a protocol largely based on *Rowan et al., 2015*, the TruSeq DNA Sample Preparation Guide from Illumina (Illumina, San Diego, CA), and *Rohland and Reich, 2012*, with the addition of an enzymatic repeats depletion step using a Duplex-Specific Nuclease (DSN; Evrogen, Moscow, Russia) (*Matvienko et al., 2013*; *Shagina et al., 2010*; *Todesco et al., 2020*). All libraries were sequenced at the McGill University and Génome Québec Innovation Center on HiSeq2500, HiSeq4000, and HiSeqX instruments (Illumina) to produce paired end, 150 bp reads.

Sequences were trimmed for low quality using Trimmomatic (v0.36) (*Bolger et al., 2014*) and aligned to the *H. annuus* XRQv1 genome (*Badouin et al., 2017*) using NextGenMap (v0.5.3) (*Sedlazeck et al., 2013*). We followed the best practice recommendations of the Genome Analysis ToolKit (GATK) (*Poplin et al., 2017*) and executed steps documented in GATK's germline short variant discovery pipeline (for GATK 4.0.1.2). During genotyping, to reduce computational time and improve variant quality, genomic regions containing transposable elements were excluded (*Badouin et al., 2017*). We then used GATK's VariantRecalibrator (v4.0.1.2) to select high-quality variants. SNP data were then filtered for minor allele frequency (MAF) $\geq$ 0.01, genotype rate $\geq$ 90%, and to keep only biallelic SNPs.

Filtered SNPs were then remapped to the improved reference assembly HA412-HOv2 (*Staton and Lázaro-Guevara, 2020*) using BWA (v0.7.17) (*Li, 2013*). These remapped SNPs were used for all analyses, excluding the GWAS for the region surrounding the *HaMYB111* locus that used unfiltered variants based on the XRQv1 assembly (*Figure 2—figure supplement 3*).

The SNP dataset used to determine the genotype at the Chr15_LUVp SNP in other species (*H. argophyllus*, *H. niveus*, *H. debilis*, *H. decapetalus*, *H. divaricatus*, and *H. grosseserratus*) was based on WGS data generated for *Todesco et al., 2020* and is described in *Owens et al., 2021*. Sequence data for the Sunflower Association Mapping population are reported in *Hübner et al., 2019*.

## Genome-wide association mapping

Genome-wide association analyses for LUVp were performed for *H. annuus, H. petiolaris petiolaris,* and *H. petiolaris fallax* using two-sided mixed models implemented in EMMAX (v07Mar2010) (*Kang et al., 2010*) or in the EMMAX module in EasyGWAS (*Grimm et al., 2017*). For all runs, the first three principal components (PCs) were included as covariates, as well as a kinship matrix. Only SNPs with MAF > 5% were included in the analyses, and variants were imputed and phased using Beagle (version 10Jun18.811) (*Browning et al., 2018* #497). A GWAS with MAF > 1% in *H. petiolaris petiolaris* failed to find any additional association between LUVp and variation at the Chr15_LUVp SNP (the L allele is found at a frequency of ~2% in *H. petiolaris petiolaris*). Sample size was estimated to be sufficient to provide an 85% probability of detecting loci explaining 5% or more of the phenotypic variance in *H. annuus*. An 85% probability of detecting loci explaining 8% of variance in *H. petiolaris* was estimated for the whole species set (488 individuals); upon analysis of resequencing data for this species, three distinct clusters of individuals were detected (*H. petiolaris petiolaris, H. petiolaris fallax, H. niveus canescens*), and GWAS were performed independently on *H. petiolaris petiolaris* and *H. petiolaris fallax* (the *H. niveus canescens* cluster included only 86 individuals). Subspecies dataset were found to provide sufficient power to detect strong associations with adaptive traits (*Todesco et al., 2020*). Narrow-sense heritability ($h^2$) in the *H. annuus* dataset was estimated using EMMAX (*Kang et al., 2010*), GEMMA (*Zhou and Stephens, 2012*), GCTA-GREML (*Yang et al., 2011*), and BOLT_REML (*Loh et al., 2015*). All software produced $h^2$ values of ~1: while it is possible that the presence of a single locus of very large effect would lead to inflation of these estimates, all individuals in the GWAS

populations were grown at the same time under uniform conditions, and limited environmental effects are therefore expected.

## F₂ populations and genotyping

Individuals from population ANN_03 from CA, USA (large LUVp), and ANN_55 from TX, USA (small LUVp), were grown in 2-gallon pots in a field. When the plants reached maturity, they were moved to a greenhouse, where several inflorescences were bagged and crossed. The resulting $F_1$ seeds were germinated and grown in a greenhouse, and pairs of siblings were crossed (wild sunflowers are self-incompatible). The resulting $F_2$ populations were grown both in a greenhouse in the winter of 2019 (n = 42 individuals for population 1, 38 individuals for population 2) and in a field as part of the 2019 common garden experiments (n = 54 individuals for population 1, 50 individuals for population 2). DNA was extracted from young leaf tissue as described above. All $F_2$ plants were genotyped for the Chr15_LUVp SNPs using a custom TaqMan SNP genotyping assay (Thermo Fisher Scientific, Waltham, MA) on a Viia 7 Real-Time PCR system (Thermo Fisher Scientific).

## Metabolite analyses

Methanolic extractions were performed following *Stracke et al., 2007*. Sunflower ligules (or portions of them) and *Arabidopsis* petals were collected and flash-frozen in liquid nitrogen. For sunflower, all ligules, or part of ligules, were collected from the selected inflorescence (avoiding damaged ligules). At least two ligules (or parts of ligules) were then randomly chosen, pooled, and weighed for methanolic extraction from each inflorescence. For *Arabidopsis*, hundreds of petals from several plants for each genotype were collected, pooled, and weighed to obtain a sufficient amount of tissue. The frozen tissue was ground to a fine powder by adding 10–15 zirconia beads (1 mm diameter) and using a TissueLyser (QIAGEN) for sunflower ligules, or using a plastic pestle in a 1.5 ml tube for *Arabidopsis* petals. 0.5 ml of 80% methanol were added, and the samples were further homogenized and incubated at 70°C for 15 min. They were then centrifuged at 15,000 × $g$ for 10 min, and the supernatant was dried in a SpeedVac (Thermo Fisher Scientifics) at 60°C. Samples were then resuspended in 1 µl (sunflower) or 2.5 µl (*Arabidopsis*) of 80% methanol for every milligram of starting tissue.

The extracts were analysed by LC/MS/MS using an Agilent 1290 UHPLC system (Agilent Technologies, Santa Clara, CA) coupled with an Agilent 6530 Quadrupole Time of Flight mass spectrometer. The chromatographic separation was performed on Atlantis T3- C18 reversed-phase (50 mm × 2.1 mm, 3 µm) analytical columns (Waters Corp, Milford, MA). The column temperature was set at 40°C. The elution gradient consisted of mobile phase A (water and 0.2% formic acid) and mobile phase B (acetonitrile and 0.2% formic acid). The gradient program was started with 3% B, increased to 25% B in 10 min, then increased to 40% B in 13 min, increased to 90% B in 17 min, held for 1 min, and equilibrated back to 3% B in 20 min. The flow rate was set at 0.4 ml/min and injection volume was 1 µl. A photo diode array (PDA) detector was used for detection of UV absorption in the range of 190–600 nm.

MS and MS/MS detection were performed using an Agilent 6530 accurate mass Quadrupole Time of Flight mass spectrometer equipped with an ESI (electrospray) source operating in both positive and negative ionization modes. Accurate positive ESI LC/MS and LC/MS/MS data were processed using the Agilent MassHunter software to identify the analytes. The ESI conditions were as follows: nebulizing gas (nitrogen) pressure and temperature were 30 psi and 325°C; sheath gas (nitrogen) flow and temperature were 12 l/min, 325°C; dry gas (nitrogen) was 7 l/min. Full scan mass range was 50–1700 *m/z*. Stepwise fragmentation analysis (MS/MS) was carried out with different collision energies depending on the compound class.

## Transgenes and expression assays

Total RNA was isolated from mature and developing ligules, or part of ligules, using TRIzol (Thermo Fisher Scientific), and cDNA was synthesized using the RevertAid First Strand cDNA Synthesis kit (Thermo Fisher Scientific). All ligules, or part of ligules, were collected from the selected inflorescence in a single tube (avoiding damaged ligules) and flash-frozen in liquid nitrogen. At least two full ligules (or parts of ligules) were then randomly chosen and pooled for RNA extraction from each inflorescence. Genomic DNA was extracted from leaves of *Arabidopsis* using CTAB (*Murray and Thompson, 1980*). A 1959-bp-long fragment (*pAtMYB111*) from the promoter region of *AtMYB111* (*At5g49330*),

including the 5'-UTR of the gene, was amplified using Phusion High-Fidelity DNA polymerase (New England Biolabs, Ipswich, MA) and introduced in pFK206 derived from pGREEN (*Hellens et al., 2000*). Alleles of *HaMYB111* (*HanXRQChr15g0465131*) were amplified from cDNA from ligules of individuals from populations ANN_03 (large LUVp, from CA) and ANN_55 (small LUVp, from TX). These are the same populations from which the parental plants of the F$_2$ populations shown in *Figure 2e* were derived. A comparison of the patterns of polymorphisms between these two alleles (*HaMYB111_large* and *HaMYB111_small*), other *HaMYB111* CDS alleles from wild *H. annuus*, and the cultivated reference XRQ sequence is shown in *Figure 3—figure supplement 2*. These alleles were placed under the control of *pAtMYB111* (in the plasmid described above) or of the constitutive *CaMV 35S* promoter (in pFK210, derived as well from pGREEN; *Hellens et al., 2000*). Constructs were introduced into *Arabidopsis* plants by *Agrobacterium tumefaciens* -mediated transformation (strain GV3101) (*Weigel and Glazebrook, 2002*). At least five independent transgenic lines with levels of UV pigmentation comparable to the ones shown in *Figure 3d* were recovered for each construct. For expression analyses, qRT-PCRs were performed on cDNA from ligules using the SsoFast EvaGreen Supermix (Bio-Rad, Hercules, CA) on a CFX96 Real-Time PCR Detection System (Bio-Rad). Expression levels were normalized against *HaEF1α*. *HaEF1α* (*HanXRQChr11g0334971*) was selected as a reference gene because, out of a set of genes that showed constitutively elevated expression across different tissues and treatments in cultivated sunflower (*Badouin et al., 2017*), it displayed the most robust expression patterns across ligules of different *H. annuus* and *H. petiolaris* individuals, and across ligule tips and bases in the two species. For the expression analyses shown in *Figure 3h and i*, portions of ligules were collected at different developmental stages from three separate inflorescences from one individual for each species (biological replicates). Three qRT-PCRs were run for each sample (technical replicates). For the expression analysis shown in *Figure 3j*, samples were collected from wild *H. annuus* individuals grown as part of the 2019 common garden experiment. Ligules were collected on the same day from developing inflorescences of 24 individuals with large LUVp (from 10 populations) and 22 individuals with small LUVp (from 11 populations). qPCRs for three technical replicates were performed for each individual. These plants were genotyped for the Chr15_LUVp SNP using a custom TaqMan assay (see 'F$_2$ populations and genotyping') on a CFX96 Real-Time PCR Detection System (Bio-Rad). Sample size for this experiment was determined by the number of available plants with opposite LUVp phenotypes and at the appropriate developmental stage on the day in which samples were collected. Primers used for cloning and qRT-PCR are given in the Key resources table.

## Sanger and PacBio sequencing

Fragments ranging in size from 1.5 to 5.5 kbp were amplified using Phusion High-Fidelity DNA polymerase (New England Biolabs) from genomic DNA of 20 individuals that had been previously resequenced (*Todesco et al., 2020*) and whose genotype at the Chr15_LUVp SNP was therefore known. Fragments were then cloned in either pBluescript or pJET (Thermo Fisher Scientific) and sequenced on a 3730S DNA analyzer using BigDye Terminator v3.1 sequencing chemistry (Applied Biosystems, Foster City, CA).

For long read sequencing, seed from wild *H. annuus* populations known to be homozygous for different alleles at the Chr15_LUVp SNP were germinated and grown in a greenhouse. After confirming that they had the expected LUVp phenotype, branches from each plant were covered with dark cloth for several days, and young, etiolated leaves were collected and immediately frozen in liquid nitrogen. High molecular weight (HMW) DNA was extracted from six plants using a modified CTAB protocol (*Stoffel et al., 2012*). All individuals were genotyped for the Chr15_LUVp SNP using a custom TaqMan SNP genotyping assay (Thermo Fisher Scientific, see above) on a CFX96 Real-Time PCR Detection System (Bio-Rad). Two individuals, one with large and one with small LUVp, were selected. HiFi library preparation and sequencing on a Sequel II instrument (PacBio, Menlo Park, CA) were performed at the McGill University and Génome Québec Innovation Center. Each individual was sequenced on an individual SMRT cell 8M, resulting in average genome-wide sequencing coverage of 6–8×.

## Pollinator preferences assays

In September 2017, pollinator visits were recorded in individual inflorescences of pairs of plants with large (from population ANN_03, from CA) and small LUVp (from population ANN_55, from TX) grown in pots in a field adjacent the Nursery South Campus greenhouses of the University of British

Columbia. Populations ANN_55 and ANN_03 were chosen because they flowered at about the same time in our 2016 common garden experiment and had inflorescences of similar size and appearance. Pairs of size-matched inflorescences, made to face towards the same direction, were filmed using a Bushnell Trophy Cam HD (Bushnell, Overland Park, KS) in 12 min intervals. Visitation rates were averaged over 14 such movies (*Figure 4—source data 1*). The only other sunflowers present in the field were *H. anomalus* individuals, grown in a separate field about 15 m away. *H. anomalus* has uniformly small floral UV patterns (*Figure 1—figure supplement 1*), and is therefore unlikely to have affected pollinator preferences.

In summer 2019, pollinator visits were scored in a common garden experiment consisting of 1484 *H. annuus* plants at the Totem Plant Science Field Station of the University of British Columbia (see the 'Common gardens' section for details on field design). Over 5 days, between 29 July and 7 August, pollinator visits on individual plants were directly observed and counted over 5 min intervals for a total of 435 series of measurements on 111 plants from 51 different populations (*Figure 4—source data 1*). Observers were careful to be at least 2 m away from the plant, and not to overshadow it. Visits to all inflorescences for each plant were recorded; pollinators visiting more than one inflorescence per plant were recorded only once. To generate a more homogenous and comparable dataset, measurements for plants with too few (1) or too many (>10) inflorescences were excluded from the final analysis (*Figure 4—source data 1*).

## Correlations with environmental variables and GEA analyses

Twenty topo-climatic factors were extracted from climate data collected over a 30-year period (1961–1990) for the geographical coordinates of the population collection sites using the software package Climate NA (*Wang et al., 2016*; *Figure 1—source data 1*). Additionally, UV radiation data were extracted from the glUV dataset (*Beckmann et al., 2014*) using the R package 'raster' (*Hijmans, 2020*; *R Development Core Team, 2020*). Correlations between individual environmental variables and LUVp was calculated using the 'lm' function implemented in R. A correlation matrix between all environmental variables, and LUVp was calculated using the 'cor' function in R and plotted using the 'heatmap.2' function in the 'gplots' package (*Warnes et al., 2009*). Plots of the interactions between relative humidity and average temperature in relation to LUVp were generated using the 'interact_plot' function implemented in the 'interactions' R package (*Long, 2020*). It should be noted that the values for climate variables used in these analyses are extrapolated from weather stations across North America, and not measured in situ, meaning that they might not account for microclimatic variation. For example, two populations in Southern Arizona do not fit the pattern we proposed – they have small floral UV patterns and high frequency of S alleles at the Chr15_LUVp SNP, despite being associated with relatively low RH values in our datasets. However, one of them (ANN_13) was collected along the Verde River, near Deadhorse lake, and the description of the collection site is 'riparian forest and wetland,' suggesting that humidity might be locally higher than in the surrounding region. Similarly, from satellite pictures, the collection site for the other population (ANN_47) appears considerably more verdant than other collection sites in Arizona.

GEAs were analysed using BayPass (*Gautier, 2015*) version 2.1. Population structure was estimated by choosing 10,000 putatively neutral random SNPs under the BayPass core model. The Bayes factor (denoted $BF_{is}$ as in *Gautier, 2015*) was then calculated under the standard covariate mode. For each SNP, $BF_{is}$ was expressed in deciban units [dB, $10 \log_{10} (BF_{is})$]. Significance was determined following *Gautier, 2015* and employing Jeffreys' rule (*Jeffreys, 1961*), quantifying the strength of associations between SNPs and variables as 'strong' ($10\ dB \leq BF_{is} < 15\ dB$), 'very strong' ($15\ dB \leq BF_{is} < 20\ dB$), and 'decisive' ($BF_{is} \geq 20\ dB$; *Figure 4—source data 4*).

## Desiccation assays

Water loss was determined by measuring changes in the weight of detached ligules and leaves over time (*Duursma et al., 2019*; *Hygen, 1951*). In the summer of 2020, fully developed inflorescences and the one or two youngest fully developed leaves from each individual were collected from well-watered, greenhouse-grown plants that had large (LUVp = 1) or small (LUVp ≤ 0.4) floral UV patterns. They were brought immediately to an environment kept at 21°C and were left overnight with their stems or petioles immersed in a beaker containing distilled water. The following morning leaves from each plant, and three ligules removed from each inflorescence (selected to be as far apart from each

other as possible across the inflorescence and taking care to avoid damaged or otherwise unrepresentative ligules), were individually weighed and hanged to air dry at room temperature (21°C). Their weight was measured at 1 hr intervals for 5 hr, and then again the following morning. Leaves and ligules were then incubated for 48 hr at 65°C in an oven to determine their dry weight. Total water content was measured as the difference between the initial fresh weight ($W_0$) and dry weight ($W_d$). Water loss was expressed as a fraction of the total water content of each organ using the formula [($W_i$-$W_d$)/($W_0$-$W_d$)] × 100, where $W_i$ is the weight of a sample at a time i. The assay was performed on ligules from 16 inflorescences from 12 individuals belonging to seven different populations of *H. annuus*, and on leaves from 15 individuals from eight different populations. Of the individuals used for assays on leaves, 10 were also used for assays on ligules, 4 were half-siblings of individuals used for ligule assays, and 1 belonged to a different population (*Figure 4—source data 3*).

## Acknowledgements

This research was conducted in the ancestral and unceded territory of the xʷməθkʷəy̓əm (Musqueam) People. We thank Andrea Todesco, Daniela Rodeghiero, Emma Borger, Quinn Anderson, Jennifer Lipka, Jasmine Lai, Hafsa Ahmed, Dominique Skonieczny, Ana Parra, Cassandra Konecny, Chris Zan, Juan Chavez, Victor Canta-Gallo, Anna Dmitrieva, Patrick Jacobsen, Kelsie Morioka, and Daniel Yang for assistance with field work and data acquisition, Melina Byron and Glen Healy at UBC, Christina Wistrom at the UC Berkeley Oxford Tract Facility, and the UC Davis Plant Sciences Field Station personnel for assistance with greenhouse and field experiments, Elizabeth Elle and Tyler Kelly for help planning the pollinator preference experiments, Laura Marek and the USDA-ARS in Ames, IA, USA, for providing sunflower seeds, and Chase Mason for providing cuttings for *Phoebantus tenuifolius*. Maps were realized using tiles from Stamen Design (https://stamen.com), under CC BY 3.0, from data by OpenStreetMaps contributors (https://openstreetmap.org), under ODbL. Funding was provided by Genome Canada and Genome BC (LSARP2014-223SUN), the NSF Plant Genome Program (IOS-1444522, IOS-1759442), the University of California, Berkeley, and an HFSP long-term postdoctoral fellowship to MT (LT000780/2013).

## Additional information

### Funding

| Funder | Grant reference number | Author |
|---|---|---|
| Genome Canada | LSARP2014-223SUN | Marco Todesco<br>Loren H Rieseberg |
| Genome British Columbia | LSARP2014-223SUN | Marco Todesco<br>Loren H Rieseberg |
| National Science Foundation | IOS-1444522 | Loren H Rieseberg |
| National Science Foundation | IOS-1759942 | Benjamin K Blackman |
| Human Frontier Science Program | LT000780/2013 | Marco Todesco |
| University of California Berkeley | | Benjamin K Blackman |

The funders had no role in study design, data collection and interpretation, or the decision to submit the work for publication.

### Author contributions

Marco Todesco, Conceptualization, Formal analysis, Funding acquisition, Investigation, Methodology, Project administration, Supervision, Visualization, Writing - original draft, Writing – review and editing; Natalia Bercovich, Ivana Imerovski, Conceptualization, Data curation, Investigation, Methodology, Project administration, Supervision, Writing – review and editing; Amy Kim, Óscar Dorado Ruiz,

Investigation; Gregory L Owens, Data curation, Formal analysis, Writing – review and editing; Srinidhi V Holalu, Formal analysis, Investigation; Lufiani L Madilao, Investigation, Methodology; Mojtaba Jahani, Investigation, Visualization; Jean-Sébastien Légaré, Data curation, Methodology; Benjamin K Blackman, Loren H Rieseberg, Conceptualization, Funding acquisition, Supervision, Writing – review and editing

**Author ORCIDs**
Marco Todesco http://orcid.org/0000-0002-6227-4096
Natalia Bercovich http://orcid.org/0000-0002-7703-2858
Amy Kim http://orcid.org/0000-0002-2623-4118
Ivana Imerovski http://orcid.org/0000-0002-1164-3664
Gregory L Owens http://orcid.org/0000-0002-4019-5215
Srinidhi V Holalu http://orcid.org/0000-0002-1948-8216
Lufiani L Madilao http://orcid.org/0000-0003-4161-2540
Mojtaba Jahani http://orcid.org/0000-0003-1844-1464
Jean-Sébastien Légaré http://orcid.org/0000-0003-1483-9643
Benjamin K Blackman http://orcid.org/0000-0003-4936-6153
Loren H Rieseberg http://orcid.org/0000-0002-2712-2417

**Decision letter and Author response**
Decision letter https://doi.org/10.7554/eLife.72072.sa1
Author response https://doi.org/10.7554/eLife.72072.sa2

## Additional files

**Supplementary files**
- Supplementary file 1. Multiple sequence alignment for *HaMYB111* coding sequence.
- Supplementary file 2. Multiple sequence alignment for *HaMYB111* genomic sequence.
- Supplementary file 3. Multiple sequence alignment for *HaMYB111* proximal promoter region.
- Supplementary file 4. Multiple sequence alignment for *HaMYB111* distal promoter region.
- Supplementary file 5. GenBank dataset details.
- Transparent reporting form

**Data availability**
All raw sequenced data are stored in the Sequence Read Archive (SRA) under BioProjects PRJNA532579, PRJNA398560 and PRJNA736734. Filtered SNP datasets are available at https://rieseberglab.github.io/ubc-sunflower-genome/. Raw short read sequencing data and SNP datasets have been previously described in (Todesco et al., 2020). The sequences of individual alleles at the *HaMYB111* locus and of *HaMYB111* coding sequences have been deposited at GenBank under accession numbers MZ597473-MZ597536 and MZ410295-MZ410296, respectively. Full details and links are provided in Supplementary file 5. All other data are available in the main text or in the source data provided with the manuscript.

The following dataset was generated:

| Author(s) | Year | Dataset title | Dataset URL | Database and Identifier |
|---|---|---|---|---|
| Todesco M, Rieseberg LH, Bercovich N, Owens GL | 2021 | Floral UV patterns in sunflowers: HiFI sequences | https://www.ncbi.nlm.nih.gov/sra/PRJNA736734 | NCBI Sequence Read Archive, PRJNA736734 |

The following previously published datasets were used:

| Author(s) | Year | Dataset title | Dataset URL | Database and Identifier |
|-----------|------|---------------|-------------|-------------------------|
| Todesco M, Rieseberg LH, Bercovich N, Owens GL, Légaré J-S | 2019 | Wild Helianthus GWAS and GEA | https://www.ncbi.nlm.nih.gov/sra/PRJNA532579 | NCBI Sequence Read Archive, PRJNA532579 |
| Todesco M, Rieseberg LH, Owens GL, Drummond EBM | 2017 | Wild and Weedy Helianthus annuus whole genome resequencing | https://www.ncbi.nlm.nih.gov/sra/PRJNA398560 | NCBI Sequence Read Archive, PRJNA398560 |

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
