## [Editor Report]

The enlarged petals of sunflowers contain pigments that absorb ultraviolet light and are perceived by pollinators as dark ‘bullseyes’ that function as nectar guides. Todesco et al. identify the primary genetic mechanism underlying variation in the size of this bullseye pattern and provide evidence suggesting that abiotic variables, rather than pollinators, may maintain this phenotypic and genotypic variation.

---

## [Decision Letter]

**Decision letter after peer review:**

Thank you for submitting your article "Genetic basis and dual adaptive role of floral pigmentation in sunflowers" for consideration by *eLife*. Your article has been reviewed by 3 peer reviewers, including Jeff Ross-Ibarra as Reviewing Editor and Reviewer #1, and the evaluation has been overseen by Jürgen Kleine-Vehn as the Senior Editor. The following individual involved in review of your submission has agreed to reveal their identity: Ian T Baldwin (Reviewer #2).

All three reviewers are quite enthusiastic about the manuscript and I believe that with some relatively straightforward revisions it would be suitable for publication in *eLife*. The reviewers have discussed their reviews with one another, and the Reviewing Editor has drafted this to help you prepare a revised submission.

Essential revisions:

The primary concern echoed by the reviewers, and which came up in our discussion, was the feeling that several of the claims of the paper could do with a bit more caveats and consideration of alternative explanations. For example, reviewer #2 points out previous results on additional metabolites that could be involved. We all agreed the claims about fitness advantages could be tempered some, given that e.g. the water loss experiment is not the same as the actual measure of transpiration, and the link between bullseye size and fitness under different humidity, while likely given other evidence in the paper, is also untested. Although we felt that additional experiments could be done to solidify these claims, we also all felt the current manuscript stands as is with the conclusions softened somewhat.

Aside from this major concern, each of the reviewers had a number of additional points to consider, and I ask you kindly respond to these in turn as well.

Thank you again for submitting such a wonderful paper to *eLife*, and I look forward to seeing the revision – I doubt very much it would need to go back out to review.

*Reviewer #1 (Recommendations for the authors):*

I'm not a fan of the last line of the conclusion. While I'm not against having some fun with papers, this felt a bit flippant perhaps.

Any speculation as to what causes the odd non-linear patterns (low UV radiation and temperature) at intermediate LUVP in figure 4c and d?

Any speculation as to why is the association with temperature stronger than RH in 4k?

I couldn't quite understand figure 3h. The bars show expression relative to what? Are these stacked bars or overlapping?

*Reviewer #2 (Recommendations for the authors):*

The putative fitness benefits of variable bullseye size under different humidity regimes, proposed to explain the observed geographical clines in bullseye size remain untested. These functional hypotheses could have been tested in accessions that naturally vary in bullseye size, or with genome-edited lines, which are subject to alternative explanations and onerous, respectively.

Alternatively, these hypotheses could have been tested by comparing the reproductive outputs under different watering regimes of myb12 Arabidopsis mutants and those complimented with the pAtMyb111 transgene; both species are largely selfing, so that seed set would be a reasonable proxy of functional significance. Water-loss associated fitness effects of floral flavonol glycoside expression in Arabidopsis would argue strongly for a generalized effect of these glycosides that does not depend on the particular molecular composition of the glycosides.

While bar graphs of pollinator visitation data from 2019 are presented in a figure, the data from 2017 are only mentioned in the figure caption of Figure 4 supplemental.

*Reviewer #3 (Recommendations for the authors):*

First, thanks for a great read! I enjoyed this manuscript.

My line notes elaborate on (and identify relevant lines for) the weaknesses I identified in my public review, and I offer possible solutions for some of them. I have only two additional comments here. (1) The authors go a bit too light on methodological details for my liking (e.g. sampling design for UV absorbance, transpiration), or leave out results that could be important and interesting (e.g. would like to Sanger sequencing variation in supplement or data). I think these should be available in the methods or a supplementary text. (2) I think there are a few areas where more could be learned (optionally!), including further exploration of GWAS, and using linear models for proportion data instead of categorizing

There are many line notes, but these are solely meant to be helpful to the authors. There are NO further critiques. Line notes are only examples, typos, or compliments.

Line 26: UV patterns are not manipulated in pollinator experiments, and they are observed across populations. Other phenotypes could vary across populations that matter to pollinators. Suggest "strongly correlate with" instead of "have a strong effect on" Same is true at line 29 about UV and transpiration.

Line 43: This is just my opinion, but the phrasing of the emoji reference feels a little informal.

Figure 1 is very easy to understand without even reading the caption. Well done. My only suggestion to emphasize the scale bar, it is a bit hard to pick out in panel c.

Line 95: This portion is a bit light on details. Here, it is not clear in R and D text what happened to ssp. fallax. Also, nowhere in results (neither Figure 1b nor text) do the authors clarify if other features (e.g. genes) are in the region that might be linked to Chr15_LUVp SNP. What is the scale of LD decay in this GWAS panel? If bigger than 30kpb, I might suggest expanding Figure 2b axis to the scale of LD breakdown. I would suggest highlighting the Chr15_LUVp SNP. Lastly, we read that from Sanger sequencing results that there are non-SNP promoter-region features and coding sequence variation, I would love to see these in the supplement.

Line 99: 62% is a lot of the variation for one SNP. I'm curious how much of the heritable variation is this? Does that increase in the greenhouse for the F2 populations?

I also wonder if there might be more to learn from the GWAS, especially for H. petiolaris? GWAS works best with a normally distributed phenotype and intermediate frequency SNPs, and these are somewhat small GWASes. Computation methods exist to account for MAF and phenotype distribution effects on spurious SNP associations (or under-associations). I doubt this would change the primary result, but might be worthwhile if the authors wish to explain more variation. See, Stanton-Geddes, John, et al., "Candidate genes and genetic architecture of symbiotic and agronomic traits revealed by whole-genome, sequence-based association genetics in Medicago truncatula." PloS one 8.5 (2013): e65688.

Line 104: how do these proportions translate for individuals where UV abs was simply reduced distally? In how many of the plants was absorbance reduced distally versus absent distally? And was this evenly distributed across populations? Did this correlate with pollinator visitation? Expression?

From line 138 onwards, individuals are referred to in phenotype bins. I imagine that the binning addresses the non-normal distribution of proportion phenotypes, but it could introduce statistical issues into the visitation and transpiration design/analyses. Maybe see: Douma, Jacob C., and James T. Weedon. "Analysing continuous proportions in ecology and evolution: A practical introduction to β and Dirichlet regression." Methods in Ecology and Evolution 10.9 (2019): 1412-1430.

Line 155: "we found flavonol glycosides to be the main UV-absorbing pigments". I see that this evidence is probably in figure 3 panels a and b, but I cannot figure out how UV-absorbance is on these graphs, nor do I understand from the methods how this was evaluated. I suggest adding detail.

Line 178: "equally effective" The figures show some differences between these alleles when paired with the local promoter. It's unclear to me what we should learn from this, please interpret. Also, which plants were selected for the large and small allele coding sequences? I can't find it in the methods. I see that variation in CDS obtained from Sanger sequencing was not associated with phenotypes, but still good to report which plants were used. Similarly, I don't understand why a myb112 mutant is explored only in the background of a myb111 mutant? I buy that MYB111 is the gene that's important regardless, just letting the authors know where they lost me if I missed something critical.

Line 196: Did HaMYB111 expression correlate with genotype at the promoter region? Did it correlate quantitatively with phenotype (beyond bins)?

Line 206: How extensive is the variation in the promoter region? Obviously across the range the GEA analysis is ideal, but within populations that have fixed/nearly fixed / maintained one allele vs the other we might expect reductions in diversity. What about other population genetic tests for selection?

Line 216: in sequencing these individuals, were the promoter regions also invariant?

Line 221: "haplotype" confuses me. I thought the S allele was a single SNP?

Line 223: Given figures in the supplement showing UV patterns across Helianthus species, I am unsure whether smaller LUVp phenotype is ancestral. The authors might run a phylogenetic analysis, or soften that claim.

Line 231: panels a, b, e: see early comment. Does peak area tells you something about amount? I'm not sure from caption/main text. Was the data for panel c generated in this study or is it from the previous study mentioned in text? For panels f, h, i, and j, the number of individuals, which genotypes/populations are not given in the caption, main text, or methods – please point to source data if they are there, and ideally also summarize. Also, for qPCR, why was HaEF1alpha chosen as the comparison?

Line 264: I disagree. Pollinators learn on short timescales. I would expect learning to play a role. For example, pollinators often display "constancy" behavior, by re-visiting a phenotype they have just visited. So if big bullseyes are common (which I expect based on figure 1), there could be a self-reinforcing bias towards visits to big bullseyes. What BC won't have is genetic variation or species filtering patterns in pollinators that might change the relative fitness of UV phenotypes across the sunflower range (though I otherwise agree with the authors' reasoning in lines 283-288). Also, can the authors comment on whether it is known if pollinators care more about proportion of UV absorbing area or the total size? Here, the authors have considered proportion only.

Line 269: This design confounds any phenotypes that are co-correlated to the UV phenotype across sunflower populations (e.g. due to local adaptation or n\eutral population structure). Therefore, pollinator preferences could arguably be due to co-varying phenotypes. Possible solution: Given that F2s within two different populations were included in the field study, was any pollinator preference data taken for these? Or are any other within population large versus small LUVp comparisons?

Line 279: I have no idea from the evidence presented how visitation links to fitness. Even the lowest rates of visitation to inflorescences seem high (>20 visits per hour, to, I assume, a limited number of florets where one visit might be sufficient).

Line 312: "explaining" is a strong word choice for a correlation.

Line 336: I have the same concern here that I do for line 269 the pollination experiment. The authors do consider and eliminate one possible co-correlated phenotype (leaf transpiration, well done) – but there are others I could think of: ligule total size, stomata density on ligules… The authors could consider the F2 individuals, or even just within population individuals, and test if this pattern is independent of population structure.

Line 339: Is there any evidence that ligules are the main source of inflorescence transpiration in sunflowers? It occurs to me that water loss might also drive pollinator preferences. Presumably pollinators avoid inflorescences with wilt – e.g. because nectar production would be low, or flowers might be old with no pollen.

Line 348-350: I am struggling to understand. The ANOVA in the source data doesn't include the fitted slopes. How does the expected effect of RH change with temperature?

Line 356: This is one result the authors interpret with the ideal amount of caution (this is a compliment). I'd be curious to know if other regions of the genome also appear in GEA, and whether any of those co-locate to other SNPs with larger estimated associations with phenotype (even if those are n.s.).

Figure 4. Why is the x-axis in 4k different from 2b? Figure 4 is otherwise functionally excellent and very aesthetically pleasing. I'm very impressed with these great figures overall, despite my critiques of Figure 3.

Conclusions: The statements on effects of pollinators and transpiration should be softened to fit the evidence. On line 390: Makes sense for Texas, but Arizona also has high S allele prevalence in Figure 2, and to my knowledge is less humid. Line 406: How is the transpiration cost of sex not about sex? I suggest deleting this sentence.

---

## [Author Response]

Reviewer #1 (Recommendations for the authors):Minor comments:I'm not a fan of the last line of the conclusion. While I'm not against having some fun with papers, this felt a bit flippant perhaps.

We have removed the last sentence.

Any speculation as to what causes the odd non-linear patterns (low UV radiation and temperature) at intermediate LUVP in figure 4c and d?

We believe that simply reflects the fact that those environmental factors (even average summer temperature, for which there is an overall good correlation with LUVp) are not the best predictors of floral UV patterns (or not as good as relative humidity). In both cases, the visible dips in the intermediate LUVp are at least partially driven from several populations with intermediate LUVp found at high latitudes (Canada, North Dakota), where UV radiation is weaker and temperatures are lower.

Any speculation as to why is the association with temperature stronger than RH in 4k?

The stronger association with temperature could hint at an effect of HaMYB111 on other traits that are important for responses to temperature, but not, or not as much, for responses to changes in relative humidity. However, in our experience the strength of associations in GEA analyses is quite sensitive to non-biological factors (using different sets of putatively neutral SNPs can affect the relative strengths of associations), and it is therefore hard to tell how much of that difference is biologically relevant. Additionally, the resulting Bayes Factors also depend on the environmental trait distribution across the landscape, so we’re cautious in overinterpreting one factor as being more important purely based on this.

I couldn't quite understand figure 3h. The bars show expression relative to what? Are these stacked bars or overlapping?

The bars in that figure are stacked bars – that was meant to recall the organization of the sunflower ligule, with the UV-absorbing part at the bottom, and the UV-reflecting part at the top. However, in hindsight that might not have been as intuitive as we had hoped. We have re-designed figures 3h and 3i to hopefully make the interpretation of the experiment more straightforward, by separating expression data for ligule tips and bases.Apologies for omitting to specify that the expression levels are normalized against the average expression levels in the UV-absorbing part of developing ligules in H. annuus. We have included that information in the figure legend, also for Figure 3j.

Reviewer #2 (Recommendations for the authors):The putative fitness benefits of variable bullseye size under different humidity regimes, proposed to explain the observed geographical clines in bullseye size remain untested. These functional hypotheses could have been tested in accessions that naturally vary in bullseye size, or with genome-edited lines, which are subject to alternative explanations and onerous, respectively.

We would have loved to be able to use transgenic or GE lines to test directly the effects of different alleles of *HaMYB111*, but unfortunately sunflower is exceptionally resistant to transformations (despite the fact that several sunflower transformation protocols have been proposed, none worked in our hand, and no functional work using those protocol has ever been published). Throughout the manuscript, we have tried to compensate for that by comparing groups of accessions with different floral UV pattern, to try to average out other factors that could differentiate individual lines. However, this would be particularly complicated for measurements of fitness, which are affected by virtually all other plant traits, meaning that all but the strongest effects would be impossible to detect (and we don’t think that variation in levels of transpiration in ligules would have an oversized effect on classic measures of plant fitness like seed set). An acceptable compromise would be to use isogenic lines in which different alleles of *HaMYB111* have been introduced into cultivated (self-compatible) sunflower. While we are working on that, obtaining suitable lines is likely to take several years.

Alternatively, these hypotheses could have been tested by comparing the reproductive outputs under different watering regimes of myb12 Arabidopsis mutants and those complimented with the pAtMyb111 transgene; both species are largely selfing, so that seed set would be a reasonable proxy of functional significance. Water-loss associated fitness effects of floral flavonol glycoside expression in Arabidopsis would argue strongly for a generalized effect of these glycosides that does not depend on the particular molecular composition of the glycosides.

While this would be a much more feasible experiment than using sunflowers, it would be complicated by the fact that *AtMYB111*, while strongly expressed in petals, is expressed also throughout the plants, and the *myb111* mutant has altered flavonol profiles in different tissues, including seedlings and rosette and cauline leaves (Stracke *et al.,* Plant J. 2007; Stracke *et al.,* New Phytol. 2010). Since accumulation of flavonol glycosides affects water loss rates in leaves/rosettes (Nakabayashi *et al.*, Plant J. 2014), as well as possibly other traits, it would be difficult to isolate the effect of floral UV pigmentation of plant fitness. Additionally, since sunflower ligules are much larger than Arabidopsis petals (even in proportion to the whole plant), the eventual effect on transpiration might be much less relevant. As mentioned above, sunflower isogenic lines will hopefully be a more suitable system to answer these questions.

While bar graphs of pollinator visitation data from 2019 are presented in a figure, the data from 2017 are only mentioned in the figure caption of Figure 4 supplemental.

We did not provide a plot with pollinator-specific information for the 2017 experiment because almost all visits were from bumblebee, with only nine visits from sylphid flies, and it would have been difficult to visualize a difference. We have now added more detailed information on the number of visits from sylphid flies in the 2017 experiment to the legend of Figure 4 —figure supplement 1, which now reads:

“In the 2017 field experiment, pollinators were overwhelmingly bumblebees. The only other pollinators recorded were syrphid flies, which visited inflorescences with large LUVp seven times (7.9% of total visits on these inflorescences) and inflorescences with small LUVp two times (3.7% of total visits on these inflorescences).” (new lines 1297-1300)

Reviewer #3 (Recommendations for the authors):First, thanks for a great read! I enjoyed this manuscript.

Thank you!

My line notes elaborate on (and identify relevant lines for) the weaknesses I identified in my public review, and I offer possible solutions for some of them. I have only two additional comments here. (1) The authors go a bit too light on methodological details for my liking (e.g. sampling design for UV absorbance, transpiration), or leave out results that could be important and interesting (e.g. would like to Sanger sequencing variation in supplement or data). I think these should be available in the methods or a supplementary text. (2) I think there are a few areas where more could be learned (optionally!), including further exploration of GWAS, and using linear models for proportion data instead of categorizingThere are many line notes, but these are solely meant to be helpful to the authors. There are NO further critiques. Line notes are only examples, typos, or compliments.Line 26: UV patterns are not manipulated in pollinator experiments, and they are observed across populations. Other phenotypes could vary across populations that matter to pollinators. Suggest “strongly correlate with” instead of “have a strong effect on” Same is true at line 29 about UV and transpiration.

We have modified the abstract as suggested on line 26. We have re-phrased line 29 to better reflect the results of our experiments (that is, that ligules with larger LUVp patterns show reduced water loss, but not directly that larger UV patterns cause reduced water loss). The two sentences now read:

“Different patterns of ultraviolet pigments in flowers are strongly correlated with pollinator preferences.” (new lines 26-27)

“Ligules with larger ultraviolet patterns, which are found in drier environments, show increased resistance to desiccation, suggesting a role in reducing water loss.” (new lines 29-31)

We have also modified slightly line 26 to include a definition of “ligules”.

Line 43: This is just my opinion, but the phrasing of the emoji reference feels a little informal.

That was our attempt to lighten the tone of the introduction, and was received with mixed review even among authors. The author that wrote the initial draft of the manuscript still finds it moderately funny, so instead of removing it we have contextualized the reference a bit better. The sentence now reads:

“Much of the popularity of sunflowers (as testified by countless references in the visual arts and, more recently, by the arguably dubious honour of being one of the only five flower species with a dedicated emoji (Unicode.org, 2020)) is due to their iconic yellow inflorescences.” (new lines 58-61)

Figure 1 is very easy to understand without even reading the caption. Well done. My only suggestion to emphasize the scale bar, it is a bit hard to pick out in panel c.

Thank you. We have increased the size bar to 2 cm, and used a thicker line.

Line 95: This portion is a bit light on details. Here, it is not clear in RandD text what happened to ssp. Fallax.

We have included some more information about the (sub-)species of wild sunflowers that are the focus of the manuscript in the previous section of RandD (“Floral UV patterns in wild sunflowers”), and we have added a mention of the fact that no significant association was found in GWAS for LUVp in *H. petiolaris fallax* (a reference to the *fallax* Manhattan plot in Figure 2 —figure supplement 1 was already present). The sentence now reads:

“While no significant association was identified for *H petiolaris fallax* (Figure 2 —figure supplement 1), we detected several genomic regions significantly associated with UV patterning in *H. petiolaris petiolaris*…” (new lines 150-152)

Also, nowhere in results (neither Figure 1b nor text) do the authors clarify if other features (e.g. genes) are in the region that might be linked to Chr15_LUVp SNP. What is the scale of LD decay in this GWAS panel? If bigger than 30kpb, I might suggest expanding Figure 2b axis to the scale of LD breakdown. I would suggest highlighting the Chr15_LUVp SNP.

We have highlighted the Chr15_LUVp SNP in Figure 2b and have include the following sentence to the legend of Figure 1b, to clarify that no other gene or annotated feature is found in the depicted interval, and that LD decays rapidly in our wild *H. annuus* panel:

“*HaMYB111* is the only annotated feature in the genomic interval shown in Figure 1b; the SNP with the strongest association to LUVp (Chr15_LUVp SNP) is highlighted in yellow. Linkage disequilibrium (LD) decays rapidly in wild *H. annuus* (average *R^2^* at 10 kbp is ~0.035 (Todesco *et al.*, 2020)), and all SNPs significantly associated with LUVp in *H. annuus* are included in the depicted region.” (new lines 184-187)

Lastly, we read that from Sanger sequencing results that there are non-SNP promoter-region features and coding sequence variation, I would love to see these in the supplement.

We have added new Figure 3 —figure supplement 2 showing the polymorphisms in the coding sequence of *HaMYB111* across a set of 18 wild *H. annuus* alleles, compared to the cultivated sunflower reference XRQ. Multiple sequence alignments for genomic regions were too large to be included as figure supplements, and contained too many polymorphisms to be meaningfully summarized in a figure similar to Figure 3 —figure supplement 2. We have provided graphical representations of these alignments as Supplementary files 1-4 (CDS, genomic *HaMYB111* region, proximal promoter region, distal promoter region), and included additional details about patterns of variation in the promoter region of *HaMYB111* in the legend of Figure 3 —figure supplement 2.

“Extensive sequence and structural variation across *H. annuus* individuals was found in intron regions as well as in the putative promoter region of *HaMYB111*. While the vast majority of the polymorphisms in introns or in the proximal promoter region (from the Chr15_LUVp SNP to the transcription start) did not appear to correlate with LUVp values or with genotypes at the Chr15_LUVp SNP, several large polymorphisms associated with both were found in the distal promoter region, upstream of the Chr15_LUVp SN. However, the distal promoter region could be amplified and sequenced in its entirety only from a subset of sunflower lines carrying the L allele at Chr15_LUVp SNP, precluding a conclusive determination of a link between this sequence variation and functional diversity between alleles of *HaMYB111*. Additionally, given that the promoter region had to be split in two large regions (proximal and distal), with limited overlap, to be amplified before Sanger sequencing, we cannot exclude the presence of more complex rearrangements in the region. Alignments for *HaMYB111* coding and genomic sequences, and for the proximal and distal promoter regions, are provided as Supplementary files 1-4.” (new lines 1275—1288)

Line 99: 62% is a lot of the variation for one SNP. I’m curious how much of the heritable variation is this? Does that increase in the greenhouse for the F2 populations?I also wonder if there might be more to learn from the GWAS, especially for H. petiolaris? GWAS works best with a normally distributed phenotype and intermediate frequency SNPs, and these are somewhat small GWASes. Computation methods exist to account for MAF and phenotype distribution effects on spurious SNP associations (or under-associations). I doubt this would change the primary result, but might be worthwhile if the authors wish to explain more variation. See, Stanton-Geddes, John, et al., "Candidate genes and genetic architecture of symbiotic and agronomic traits revealed by whole-genome, sequence-based association genetics in Medicago truncatula." PloS one 8.5 (2013): e65688.

As mentioned in one of our replies to Reviewer 1, heritability is extremely high for LUVp in our *H. annuus* GWAS population (~1, as calculated by four separate software) – meaning that the percentage of phenotypic variation explained by the Chr15_LUVp SNP is the same as the percentage of additive variation explained by that SNP.

We ran GWA analyses for *H. annuus* with a different software (GEMMA, Zhou and Stephens, Nat Genet. 2012), which has sometimes produced clearer associations in our hands, but results were virtually indistinguishable. Although other approaches might identify better associations for the *H. petiolaris* datasets, we suspect that their limited sample size (due to the separation between subspecies) would allow only limited power to detect most associations. While exploring more in details the genetics of floral UV patterns in other sunflower species is of course of interest to us, we preferred therefore to focus on *H. annuus* in the present manuscript.

Line 104: how do these proportions translate for individuals where UV abs was simply reduced distally?

We found that accounting for partial UV-absorbance in the distal part of the ligule improved the strength of the association between LUVp and *HaMYB111*, but did not affect the overall GWA pattern. Ignoring that partial pigmentation (“unmodified LUVp”) resulted in only minor reductions in average LUVp values for different genotype classes for the Chr15_LUVp SNP (L/L = 0.78 -> 0.73; L/S 0.59 -> 0.55; S/S 0.33 ->0.32). We have added this information to the methods section, and included the unmodified LUVp measurements for both individuals and genotypic classed in Figure 1 – source data 2.

In how many of the plants was absorbance reduced distally versus absent distally? And was this evenly distributed across populations?

Excluding plants with completely UV-absorbing ligules, some degree of UV-absorbance in the distal part of ligules was found on ~43% of individuals. Distal UV absorbance was generally more common in ligules with larger unmodified LUVp, and rarer in plants with small unmodified LUVp values (see Author response image 1); however, it was segregating in most populations with intermediate unmodified LUVp values.

**Author response image 1. sa2fig1:** 

Did this correlate with pollinator visitation? Expression?

In all experiments, plants in the “small” or “intermediate” LUVp classes were selected to be without noticeable distal UV absorption, to simplify data interpretation.

The corresponding methods section was amended to provide additional information on this:

“Partial UV absorbance in the tip of ligules was more common in plants with larger floral UV patterns; while accounting for this in LUVp measurements (as outline above) improved the strength of the association with the Chr15_LUVp SNP in GWAS (from *P* = 8.52e^-19^ to *P* = 5.81e^-25^), it did not change the overall pattern. Similarly, ignoring UV absorbance in the tip of ligules had only a minor effect on the average LUVp values for genotypic classes at the Chr15_LUVp SNP (Figure 1 – source data 2). To avoid possible confounding effects, for all experiments plants in the “small” and “intermediate” LUVp classes were selected to have no noticeable UV absorbance in the tips of ligules.” (new lines 644-651 Partial UV absorbance in the tip of ligules)

From line 138 onwards, individuals are referred to in phenotype bins. I imagine that the binning addresses the non-normal distribution of proportion phenotypes, but it could introduce statistical issues into the visitation and transpiration design/analyses. Maybe see: Douma, Jacob C., and James T. Weedon. "Analysing continuous proportions in ecology and evolution: A practical introduction to β and Dirichlet regression." Methods in Ecology and Evolution 10.9 (2019): 1412-1430.

As mentioned in our response to Reviewer 1, experiments were designed to compare defined phenotypic classes to reduce experimental noise and simplify interpretation. As a consequence, plants with LUVp values falling outside of these categories (e.g. 0.3 < LUVp < 0.5 and 0.8 < LUVp < 0.95 in the 2019 pollinator visitation experiment) are not represented. Therefore, we believe that analyzing these experiments in a different way than they were designed would be more problematic. However, in the revised manuscript we have provided a modified Figure 4 —figure supplement 1 in which individual data points are show (colour-coded by pollinator type), as well as a fitted lines showing the general trend across the data. We have also modified the main text to clarify that plants were purposely selected to belong to those three phenotypic classes:

“We selected plants falling into three categories of LUVp values, representatives of the more abundant phenotypic classes across the range of wild *H. annuus* (Figure 1d): small (LUVp = 0-0.3); intermediate (LUVp = 0.5-0.8) and large (LUVp > 0.95).” (new lines 361-370)

Line 155: "we found flavonol glycosides to be the main UV-absorbing pigments". I see that this evidence is probably in figure 3 panels a and b, but I cannot figure out how UV-absorbance is on these graphs, nor do I understand from the methods how this was evaluated. I suggest adding detail.

Apologies for the lack of details. We have now specified in the legend of panels 3a,b that the areas of the peaks in the chromatograms are proportional to the total UV absorbance at 350 nm for the corresponding compound in each extract. We have also included the full names of the compounds in the legend of those panels, as well as in that of Figure 3e, and specified in the text that quercetin glycosides are the main flavonols in sunflower ligules (as well as addressing the presence of caffeoyl quinic acid, CQA, in ligules).

The statement that flavonol glycosides are the main UV-absorbing pigment was based on the fact that the total area under the corresponding peaks is considerably larger than that of CQA in UV-absorbing (parts of) ligules. We have modified the sentence in line 155 to better qualify this point:

“Analysis of sunflower ligules found two main UV-absorbing compounds: glycoside conjugates of quercetin (a flavonol) and di-O-caffeoyl quinic acid (CQA, a member of a family of antioxidant compounds that includes chlorogenic acid and that accumulates at high levels in many sunflower tissues (Koeppe *et al.*, 1970)). Both quercetin glycosides and CQA were more abundant at the base of sunflower ligules, and in ligules of plants with larger LUVp. However, this pattern was much more dramatic for flavonols, and they explained a much larger fraction of the total UV absorbance in UV-absorbing (parts of) ligules, suggesting that flavonols are the main pigments responsible for UV patterning in sunflower ligules (Figure 3a,b)” (new lines 219-227)

Line 178: "equally effective" The figures show some differences between these alleles when paired with the local promoter. It's unclear to me what we should learn from this, please interpret.

We modified Figure 3d to show, for each transgenic line, a petal from three independent primary transformation events representing the range of variation observed for that line, rather than three petals from a single line. Hopefully this better represents that lack of obvious differences in the ability of different *HaMYB111* alleles in complementing the *myb111* mutant.

Also, which plants were selected for the large and small allele coding sequences? I can't find it in the methods. I see that variation in CDS obtained from Sanger sequencing was not associated with phenotypes, but still good to report which plants were used.

The large and small allele used in those complementation experiments were cloned from individuals from the same wild populations as the parental lines of the F_2_s shown in Figure 2e: ANN_03, from California (*HaMYB111_large*) and ANN_55, from Texas (*HaMYB111_small*). We have added these sequences to the new Figure 3 —figure supplement 2, where they are compared to other *HaMYB111* alleles from wild *H. annuus* individuals, as well as to the cultivated reference XRQ sequence. We have included this information in the Methods section:

“Alleles of *HaMYB111* (*HanXRQChr15g0465131*) were amplified from cDNA from ligules of individuals from populations ANN_03 (large LUVp, from California) and ANN_55 (small LUVp, from Texas). These are the same populations from which the parental plants of the F_2_ populations shown in Figure 2e were derived. A comparison between the patterns of polymorphisms between these two alleles (*HaMYB111_large* and *HaMYB111_small*), other *HaMYB111* CDS alleles from wild *H. annuus*, and the cultivated reference XRQ sequence is shown in Figure 3 —figure supplement 2.” (new lines 791-797)

Similarly, I don't understand why a myb112 mutant is explored only in the background of a myb111 mutant? I buy that MYB111 is the gene that's important regardless, just letting the authors know where they lost me if I missed something critical.

We originally provided visible and UV pictures for several lines (including *myb12*) in Figure 3 —figure supplement 1, but we have now incorporated all the content of that figure supplement in Figure 3d. That figure now shows that the *myb12* mutant by itself has no visible effect on petal UV absorbance, and its effect is only visible in conjunction with the *myb111* mutant. Similarly, the *myb111/myb12* mutant was added to Figure 3e only to show that, while *AtMYB12* does contribute to flavonol accumulation in petals, its contribution is negligible compared to that of *AtMYB111*.

Line 196: Did HaMYB111 expression correlate with genotype at the promoter region? Did it correlate quantitatively with phenotype (beyond bins)?

The strength of the correlation is largely unchanged when *HaMYB111* expression levels are analyzed based on LUVp categories (*R^2^* = 0.123, *P* = 0.0097), LUVp values (*R^2^* = 0.121, *P* = 0.0102), or genotypes at the Chr15_LUVp SNP (*R^2^* = 0.088, *P* = 0.0282). Like for the pollinator experiments, plants in the two LUVp categories were chosen to have very divergent LUVp values (small: LUVp <0.35; large: LUVp > 0.8), which explains why the correlation is almost unchanged when using LUVp values or LUVp categories.

As mentioned in the manuscript, collecting at exactly the same ligule developmental stage across a set of 46 wild lines is nearly impossible. Together with the strong dependency of *HaMYB111* expression on developmental stage, this resulted in rather noisy data, which could explain the relatively weak correlation coefficients.

We have added to the manuscript a new source data file (Figure 3 – source data 2), which includes expression data for Figures 3c,f,h,i,j, as well as new Chr15_LUVp SNP genotype data for the plants in Figure 3j (determined using a custom TaqMan assay, see Methods section). We have added information about the strength of correlation with LUVp values and Chr15_LUVp SNP genotype to the legend of Figures 3j and in Figure 3 – source data 2.

Line 206: How extensive is the variation in the promoter region? Obviously across the range the GEA analysis is ideal, but within populations that have fixed/nearly fixed / maintained one allele vs the other we might expect reductions in diversity. What about other population genetic tests for selection?

The presence of numerous large indels in the regions makes it difficult to provide a more precise quantification of the level of divergence in the region; however, to provide that information, we added graphic representations of alignments for the promoter regions of up to 15 *H. annuus* alleles (based on Sanger sequencing data) as Supplementary files 3 and 4.

SNP data in the *HaMYB111* promoter region is sparse, since several portions of it were deemed too repetitive to be used for short read mapping (see large regions with no SNPs in Figure 2b and Figure 4k), and Sanger sequencing made it clear that those SNPs provide a very limited representation of the diversity present in the region. As a consequence, we believe selection tests done using those SNP data would not be informative.

Line 216: in sequencing these individuals, were the promoter regions also invariant?

The promoter regions that we sequenced for *H. argophyllus* and *H. petiolaris* had several polymorphisms that differentiated them from *H. annuus* alleles. However, they all were overall more similar to promoter regions of individuals carrying the S allele at Chr15_LUVp than to individuals carrying the L allele.

We have added that information to the main text, and provided alignments comparing the promoter region in *H. annuus*, *H. argophyllus* and *H. petiolaris* individuals as Supplementary files 3 and 4.

The modified sentence now reads:

“Interestingly, when we sequenced the promoter region of *HaMYB111* in several *H. argophyllus* and *H. petiolaris* individuals, we found that they all carried the S allele at the Chr15_LUVp SNP, and that their promoter regions were generally more similar in sequence to those of *H. annuus* individuals carrying the S allele at the Chr15_LUVp SNP (Supplementary files 3,4).” (new lines 288-291)

Line 221: "haplotype" confuses me. I thought the S allele was a single SNP?

We have removed that part of the sentence, ad have clarified that the similarities between *HaMYB111* alleles from *H. annuus* individuals with small LUVp, and *H. argophyllus* and *H. petiolaris* extends to the whole promoter region (see above).

Line 223: Given figures in the supplement showing UV patterns across Helianthus species, I am unsure whether smaller LUVp phenotype is ancestral. The authors might run a phylogenetic analysis, or soften that claim.

In that sentence we were referring specifically to the S allele at the Chr15_LUVp SNP, and by extension to the associated *HaMYB111* allele, rather than to the small LUVp phenotype (other alleles or genes might be responsible for large UV patterns in other sunflower species). However, we agree that this was not very clear and not a particularly well-supported claim, so we have removed it.

Line 231: panels a, b, e: see early comment. Does peak area tells you something about amount? I'm not sure from caption/main text. Was the data for panel c generated in this study or is it from the previous study mentioned in text? For panels f, h, i, and j, the number of individuals, which genotypes/populations are not given in the caption, main text, or methods – please point to source data if they are there, and ideally also summarize. Also, for qPCR, why was HaEF1alpha chosen as the comparison?

We have clarified in the legend of the figure that the peaks area in those UV chromatograms is proportional to the total absorbance for that compound in the extract. We have also specified that the data used for Figure 3c and 3f were obtained from previous publications.

We have added a new Figure 3 – source data 2 file containing the expression data that Figure 3c,f,h,i,j are based on, as well as the IDs, populations of origin, LUVp values and genotype at the Chr15_LUVp SNP of the individuals used for the experiment in Figure 3j. We have expanded the legend and Methods section to provide more details about the qPCR experiments, and to clarify how HaEF1alpha was chosen as reference gene.

“*HaEF1α* (*HanXRQChr11g0334971*) was selected as a reference gene because, out of a set of genes that showed constitutively elevated expression across different tissues and treatments in cultivated sunflower (Badouin et al., 2017), it displayed the most robust expression patterns across ligules of different *H. annuus* and *H. petiolaris* individuals, and across ligules tips and bases in the two species.” (new lines 805-809)

Line 264: I disagree. Pollinators learn on short timescales. I would expect learning to play a role. For example, pollinators often display "constancy" behavior, by re-visiting a phenotype they have just visited. So if big bullseyes are common (which I expect based on figure 1), there could be a self-reinforcing bias towards visits to big bullseyes. What BC won't have is genetic variation or species filtering patterns in pollinators that might change the relative fitness of UV phenotypes across the sunflower range (though I otherwise agree with the authors' reasoning in lines 283-288). Also, can the authors comment on whether it is known if pollinators care more about proportion of UV absorbing area or the total size? Here, the authors have considered proportion only.

The experiment shown in Figure 4a compared individual plants with either very large or very small UV patterns (the inflorescences shown in Figure 4e are actually from that experiment). Only four plants from each of these two classes were present in the field. The only other sunflowers in the vicinity (~10-20 meters away) were a couple dozen *H. anomalus* plants, which have uniformly small LUVp values (see Figure 1 —figure supplement 1). While a learned bias toward large UV patterns in sunflowers would therefore be unlikely in this case, we agree that sentence might over-simplify the situation, and have removed it.

Throughout the manuscript, we have focused on UV proportions because it has been shown to be a more robust way to quantify UV patterns than the total size of UV bullseyes, at least in sunflowers (Moyers *et al.,* Ann Bot 2017), and because it is the standard in the field. While differences in inflorescence size clearly have a genetic basis, there can be large variation in size between inflorescences of a same plant, depending on developmental stage or environmental factors (and this will affect total UV size). While it is possible that total UV size would affect pollinator preferences, we do not therefore have sufficient data to test that; it should be noted, however, that the plants used for the experiment in Figure 4a were selected to have matching flowering time and inflorescence size, and the individual inflorescences that were recorded were size-matched.

We have added the information reported above in the legend of Figure 4 and in the relevant parts of the Methods section, where we have also added additional information about the experimental setting for pollinator preference assays.

Line 269: This design confounds any phenotypes that are co-correlated to the UV phenotype across sunflower populations (e.g. due to local adaptation or neutral population structure). Therefore, pollinator preferences could arguably be due to co-varying phenotypes. Possible solution: Given that F2s within two different populations were included in the field study, was any pollinator preference data taken for these? Or are any other within population large versus small LUVp comparisons?

We did not record pollinator preferences in F_2_ populations, and most wild populations have relatively uniform LUVp values. We have added a sentence acknowledging that presence of other, unmeasured traits that could also affect the pollinator preference patterns we observed. However, we believe that the fact that our results are consistent with what is reported in literature for the effect of UV bullseyes on pollinator preferences in other species supports our interpretation.

The section now reads:

“Therefore, we monitored pollinator visitation in plants grown in a common garden experiment including 1484 individuals from 106 *H. annuus* populations, spanning the entire range of the species. Assaying a much more diverse population of *H. annuus* individuals should reduce effects of traits unrelated to floral UV pigmentation on pollinator preferences” (new lines 356-359).

Line 279: I have no idea from the evidence presented how visitation links to fitness. Even the lowest rates of visitation to inflorescences seem high (>20 visits per hour, to, I assume, a limited number of florets where one visit might be sufficient).

Unfortunately, accurately measuring the effect of pollinator rates on fitness (i.e. seed set) would have required that we constantly measure pollinator visits to individual inflorescences, and to have hand pollinated inflorescences as a comparison, which was not feasible in our experimental setup. However, it has been shown before that pollination rates are yield-limiting in hybrid sunflower production (Greenleaf *et al.*, PNAS 2006). We have included this information and softened our claims about selection and fitness in this section, which now reads:

“Pollination rates are known to be yield-limiting in sunflower (Greenleaf et al., 2006), and a strong reduction in pollination could therefore have a negative effect on fitness; this would be consistent with the observation that plants with very small LUVp values were rare (~1.5% of individuals) in our common garden experiment, which was designed to provide a balanced representation of the natural range of *H. annuus*” (new lines 373-379)

More in generally, there is ample literature showing that pollinator visits increase seed yield even in self-compatible cultivated sunflower, albeit with considerable differences in effect size (e.g. Dag *et al.*, Am. Bee J. 2002; Nderitu *et al.*, Span. J. Agric. Res. 2008; Mallinger and Prasifka, Crop Sci. 2017; Said *et al.*, Pak. J. Zool. 2017; but see also Bartual *et al.*, 2018 PLoS one and Astiz *et al.*, Helia 2011); however, it is of course harder to translate this information to wild sunflowers.

Line 312: "explaining" is a strong word choice for a correlation.

Changed to “…with lower average summer temperatures being associated with larger LUVp values in *H. annuus*.” (new lines 422-423)

Line 336: I have the same concern here that I do for line 269 the pollination experiment. The authors do consider and eliminate one possible co-correlated phenotype (leaf transpiration, well done) – but there are others I could think of: ligule total size, stomata density on ligules… The authors could consider the F2 individuals, or even just within population individuals, and test if this pattern is independent of population structure.

As for the pollination experiments, we chose to focus on a diverse collection of wild genotypes because they are more representative of phenotypic variation found in the wild, but we recognize that some confounding factors could persist, and have acknowledged that in the revised manuscript. In this case as well, however, we think that the presence of independent evidence in the literature on the role of flavonol glycosides in limiting desiccation/transpiration lends support to our interpretation.

The last sentence of this section now reads:

“While desiccation rates are only a proxy for transpiration in field conditions (Duursma et al., 2019, Hygen et al., 1951), and other factors might affect ligule transpiration in this set of lines, this evidence (strong correlation between LUVp and summer relative humidity; known role of flavonol glycosides in regulating transpiration; and correlation between extent of ligule UV pigmentation and desiccation rates) suggests that variation in floral UV pigmentation in sunflowers is driven by the role of flavonol glycosides in reducing water loss from ligules, with larger floral UV patterns helping prevent drought stress in drier environments.” (new lines 462-469)

Line 339: Is there any evidence that ligules are the main source of inflorescence transpiration in sunflowers? It occurs to me that water loss might also drive pollinator preferences. Presumably pollinators avoid inflorescences with wilt – e.g. because nectar production would be low, or flowers might be old with no pollen.

To our knowledge, the amount of transpirations from different parts of the sunflower inflorescence has not been measured – however, given that they are the largest exposed surface on the inflorescence, and have a high surface-to-volume ration, it seems plausible that they represent a sizable part of the total transpiration from inflorescences.

It also seems plausible that pollinators might not like inflorescences with wilted ligules, as the reviewer suggests, since they might be old or sickly, and in an earlier version of the manuscript we had a sentence suggesting as much. However, we decided to remove it, since we had no direct evidence of that in sunflowers (although we do know that pollinators do not like inflorescences without ligules, see legend of Figure 4 —figure supplement 3).

Line 348-350: I am struggling to understand. The ANOVA in the source data doesn't include the fitted slopes. How does the expected effect of RH change with temperature?

We have included new panels (Figure 4 —figure supplement 2b; and Figure 4 —figure supplement 4e) that show how the correlation between temperature and LUVp changes at different levels of relative humidity in *H. annuus* and *H. petiolaris*. For *H. annuus* in particular, there is a strong interaction between the two variables, with a stronger negative correlation between temperature and LUVp at higher values of relative humidity.

Line 356: This is one result the authors interpret with the ideal amount of caution (this is a compliment). I'd be curious to know if other regions of the genome also appear in GEA, and whether any of those co-locate to other SNPs with larger estimated associations with phenotype (even if those are n.s.).

GEAs generally have many significant associations throughout the genome (see Todesco *et al.*, Nature 2020 for some examples with the same datasets used in this manuscript); this is because climate variables like temperature and relative humidity have major effects on many aspects of plant development and plant fitness, and elicit complex adaptations. We have not found any major GEA signal linked to other significant or suggestive GWAS associations; however, it is likely that a larger population size would be required to reliably detect weaker signals in GWA experiments.

Why is the x-axis in 4k different from 2b?

The choice of the region shown on the x-axis was arbitrary, but in both cases we wanted to show the “shape” of the association (that is, that the strongest association in the region was on top of *HaMYB111*); since the peak is broader for GEA, we included a larger region.

Figure 4 is otherwise functionally excellent and very aesthetically pleasing. I'm very impressed with these great figures overall, despite my critiques of Figure 3.

Thank you! Hopefully the changes we made to Figure 3 have addressed the Reviewer’s concerns.

Conclusions: The statements on effects of pollinators and transpiration should be softened to fit the evidence. On line 390: Makes sense for Texas, but Arizona also has high S allele prevalence in Figure 2, and to my knowledge is less humid.

It is correct that RH values for Arizona are generally lower than for Texas, and we do not mean to claim a perfect correlation between temperature/RH and floral UV patterns. It should be noted, however, that the RH values that we used for our analyses are extrapolated from weather stations across North America, and not measured *in situ*, meaning that they do not account for microclimatic variation. Of the two populations in Arizona that have smaller floral UV patterns and high frequency of S alleles, one (ANN_13) was collected along the Verde River, near Deadhorse lake, and the description of the collection site is “riparian forest and wetland”, suggesting that humidity might be locally higher than in the surrounding region. We have added this observation to the Methods section:

“It should be noted that the values for climate variables used in these analyses are extrapolated from weather stations across North America, and not measured *in situ*, meaning that they might not account for microclimatic variation. For example, two populations in Southern Arizona do not fit the pattern we proposed – they have small floral UV patterns and high frequency of S alleles at the Chr15_LUVp SNP, despite being associated with relatively low RH values in our datasets. However, one of them (ANN_13) was collected along the Verde River, near Deadhorse lake, and the description of the collection site is “riparian forest and wetland”, suggesting that humidity might be locally higher than in the surrounding region. Similarly, from satellite pictures, the collection site for the other population (ANN_47) appears considerably more verdant than other collection sites in Arizona.“ (new lines 881-892)

As suggested, we have softened the conclusions in regard to the connection between floral UV patterns and transpiration, and added a sentence to explore possible additional factors affecting pollinator preferences and/or geographic distribution of floral UV patterns.

The relevant section now reads:

“Here, we show that regulatory variation at a single major gene, the transcription factor *HaMYB111*, underlies most of the diversity for floral UV patterns in the common sunflower, wild *H. annuus*. Variation for these floral UV patterns correlates strongly with pollinator preferences, but also with geoclimatic variables (especially relative humidity and temperature) and desiccation rates in sunflower ligules. While the effects of floral UV patterns on pollinator attraction are well-known, these associations suggest a role of environmental factors in shaping diversity for this trait. Larger floral UV patterns, due to accumulation of flavonol glycoside pigments in ligules, could help reduce the amount of transpiration in environments with lower relative humidity, preventing excessive water loss and maintaining ligule turgidity. In humid, hot environments (e.g. Southern Texas), lower accumulation of flavonol glycosides would instead promote transpiration from ligules, keeping them cool and avoiding overheating.” (new lines 527-537)

Line 406: How is the transpiration cost of sex not about sex? I suggest deleting this sentence.

We have removed the sentence.